# Genomic variation of European beech reveals signals of local adaptation despite high levels of phenotypic plasticity

Desanka Lazic [1], Cornelia Geßner [1], Katharina J. Liepe [1], Isabelle Lesur-Kupin[2], Malte Mader [1], Céline Blanc-Jolivet[1], Dušan Gömöry [3], Mirko Liesebach[1], Santiago C. González-Martínez [2], Matthias Fladung [1], Bernd Degen [1] & Niels A. Müller [1] ✉

Local adaptation is key for ecotypic differentiation and species evolution. Understanding underlying genomic patterns can allow the prediction of future maladaptation and ecosystem stability. Here, we report the whole-genome resequencing of 874 individuals from 100 range-wide populations of European beech (*Fagus sylvatica* L.), an important forest tree species in Europe. We show that genetic variation closely mirrors geography with a clear pattern of isolation-by-distance. Genome-wide analyses for genotype-environment associations (GEAs) identify relatively few potentially adaptive variants after correcting for an overwhelming signal of statistically significant but non-causal GEAs. We characterize the single high confidence genomic region and pinpoint a candidate gene possibly involved in winter temperature adaptation via modulation of spring phenology. Surprisingly, allelic variation at this locus does not result in any apparent fitness differences in a common garden. More generally, reciprocal transplant experiments across large climate distances suggest extensive phenotypic plasticity. Nevertheless, we find indications of polygenic adaptation which may be essential in natural ecosystems. This polygenic signal exhibits broad- and fine-scale variation across the landscape, highlighting the relevance of spatial resolution. In summary, our results emphasize the importance, but also exemplify the complexity, of employing natural genetic variation for forest conservation under climate change.

Terrestrial plants and specifically trees make up the majority of the Earth's biomass[1,2], thereby giving forests an essential role for global carbon fluxes. With storage potentials reaching up to 25% of the current atmospheric carbon pool, forest restoration might represent one of the most effective strategies for climate change mitigation[3,4]. However, rather than being restored to mitigate climate change, forests are themselves severely threatened by rapidly changing climates and may experience marked habitat reductions[5,6]. To better understand and counteract this threat, there is an urgent need for reliable estimates of species' responses to climate change. Such estimates can be derived from species distribution models, which are based on current climate limits. However, these models treat species as homogenous and static units and do not account for adaptive differentiation and the variable adaptive capacity of local populations[7,8]. The increasing feasibility of generating large population genomic data to elucidate the genetic basis of local adaptation, and combining these data with reciprocal transplant and common garden experiments calls for an integration of intraspecific genetic variation into conservation biology frameworks[9–11].

[1]Thünen Institute of Forest Genetics, Grosshansdorf, Germany. [2]BIOGECO, INRAE, University of Bordeaux, Cestas, France. [3]Faculty of Forestry, Technical University in Zvolen, Zvolen, Slovakia. ✉e-mail: niels.mueller@thuenen.de

To elucidate the genetic basis of local adaptation, one straightforward approach is to perform genotype-environment association (GEA) analyses. For this, all that is needed in principle is genomic data for a sufficiently large number of georeferenced individuals adapted to different environments, although additional factors, such as confounding population structure or the expected genetic complexity, should be considered[12,13]. By associating each sequence variant in the genome with the relevant environmental variables, loci potentially involved in adaptation to differences in those variables can be identified. These data can then be integrated into prediction models of species' responses to climate change. The usefulness of GEAs for such predictions has already been demonstrated for conservation biology, e.g., for predicting vulnerability of coral reefs to heatwaves[14].

Additionally, genomic and environmental information can be integrated to estimate 'genomic offset', which is a proxy for potential future maladaptation and has been empirically demonstrated to outperform climate distance models[15,16]. Despite inherent uncertainties[17–19], such as the impact of different training sets[20] or the specific relationship between local adaptation and future climate vulnerability[21], genomic offset estimates could contribute to guiding future breeding and conservation efforts. This has been exemplified for animals and plants, such as the yellow warbler, a migratory songbird in North America[22], or pearl millet, a cereal crop in West Africa[23]. But perhaps the most striking possible application for genomic offset estimates can be envisioned for long-lived organisms such as trees. Due to their long juvenile phase and exceedingly long generation times, the fitness of populations cannot easily be experimentally tested under different environmental conditions. If, however, the natural sequence variants associated with adaptation to specific environmental conditions were known, future performance of populations could be predicted based on the gap between the current and the required future genetic make-up. Populations that appear identical today may exhibit markedly different levels of maladaptation in the future. It should be noted though that this not only requires the presence of (pre)adaptive genetic differentiation between populations but also a precise knowledge of the environmental drivers of potential future population decline.

In Central Europe, the ecologically and economically most important broadleaf species is European beech (*Fagus sylvatica* L.), henceforward referred to as beech. It represents a prime example of a long-lived species likely affected by future climate maladaptation. Currently, beech still represents the dominant species of the potential natural vegetation (PNV) in many European countries[24]. Its wide distribution range extends from Spain in the southwest, to Sweden in the north all the way to Greece and Bulgaria in the southeast, highlighting its huge ecological amplitude. Accordingly, evidence of local adaptation has been reported at different geographical scales[25,26], although phenotypic plasticity appears pervasive[27,28]. Tree ring data indicate that beech may experience growth rate declines of 20–50% in 70 years from now[8]. However, these predictions do not consider natural genetic variation and thus ignore possible differences in the adaptive potential of different local populations.

Here, to gain a detailed understanding of the genetic architecture of local adaptation and potential patterns of maladaptation under future climates, we characterize genomic variation of beech across its distribution range and relate it to tree performance in two common gardens. We show that genetics closely mirror geography and provide first insights into the genetic basis of local adaptation. However, missing heritability appears to preclude the identification of large parts of the adaptive variation. Interestingly, potential future maladaptation exhibits broad- and fine-scale variation across the landscape, emphasizing the importance of spatial resolution for conservation genomics studies.

## Results
### Genomic variation mirrors geography
To sample the range-wide genetic diversity of beech, we took advantage of a common garden being comprised of 100 populations from across the range (Supplementary Fig. 1, Supplementary Data 1). We randomly selected nine trees per population for whole-genome resequencing with an average sequencing depth of 42.5x (Supplementary Data 2). Mapping the resequencing data to the chromosome-level beech reference genome[29], we identified a total of 3.68 million high-confidence sequence variants, that is SNPs and short indels, of which about 540 thousand were largely independent, exhibiting linkage disequilibrium (LD) values below 0.2. Analysis of these independent variants revealed the presence of close relatives, such as half sibs or first cousins, in some populations (Supplementary Fig. 2). The material used for the establishment of the common garden thus only involved a limited number of seed trees in some cases. To avoid any artefacts in our subsequent analyses due to kinship structure, we removed individuals with pairwise relatedness of 2nd degree or higher. Additionally, we excluded one genetically highly divergent population from Bulgaria, potentially representing a hybrid with the sister species *F. orientalis* (Lipsky), and one apparently admixed population from Germany (Supplementary Fig. 3). This left us with a final set of 653 individuals from 98 populations (Supplementary Data 3).

Strikingly, a two-dimensional visualization of the genomic variation of these 653 individuals by principal component analysis (PCA) revealed a close correspondence with the map of Europe (Fig. 1a). Especially individuals from the western part of the range, that is Spain, France, Germany, Denmark and Sweden, exhibited a remarkable correlation between the first two principal components (PCs) and geography. The relationship between PC1 and longitude was high across all populations with an adjusted $R^2$ of 0.91 (Fig. 1b, d). PC2, however, varied mostly by latitude (adjusted $R^2 = 0.66$, Fig. 1c, e). Marked geographical structure can be observed up to PC6 (Supplementary Fig. 4).

### Population genomics reveals three ancestral genetic clusters and isolation-by-distance
Analysis of the individuals' ancestry coefficients using 'sparse non-negative matrix factorization' (snmf)[30] indicated the presence of three ancestral genetic clusters and substantial admixture (Fig. 2a, Supplementary Fig. 5). The cluster with the highest number of individuals in our dataset dominates the central part of the distribution range while the other two ancestry clusters are mainly found in the east and in the west, respectively (Fig. 2a, b). Whereas the genetic diversity is similar among clusters ($\pi = 1.66e$-3 to 1.73e-3) genetic differentiation is highest between the more distant eastern and western groups with an $F_{ST}$ value of 0.040 compared to 0.029 (central vs. west) and 0.016 (central vs. east) (Fig. 2c). Despite the generally low levels of genetic differentiation, the populations exhibit a highly significant signal of isolation-by-distance (Mantel test $r = 0.79$, $p < 0.001$) which becomes especially apparent due to the large geographical extent of our sampling (Fig. 2d).

Together, these results demonstrate a high level of accuracy of the common garden used for our analyses, as any mistakes in the organization or planting of the seeds and seedlings would have resulted in outliers or an overall erosion of the geographic signal in the genomic data. Additionally, the results show the absence of any pronounced human impact by seed transfer on the genetic composition of our populations. All populations, except a single one from the southern Czech Republic (showing as an outlier in Fig. 1d, e or Fig. 2b), appear largely autochthonous and can therefore be considered representative of natural populations that have evolved under the environmental conditions of their origins. Considering the large environmental amplitude of the populations sampled, relatively strong selection for adaptive differentiation may be expected. The autochthony of the populations together with the broad environmental range open the exciting possibility of identifying genotype-environment associations (GEAs) that can provide insights into the genetic basis of local adaptation.

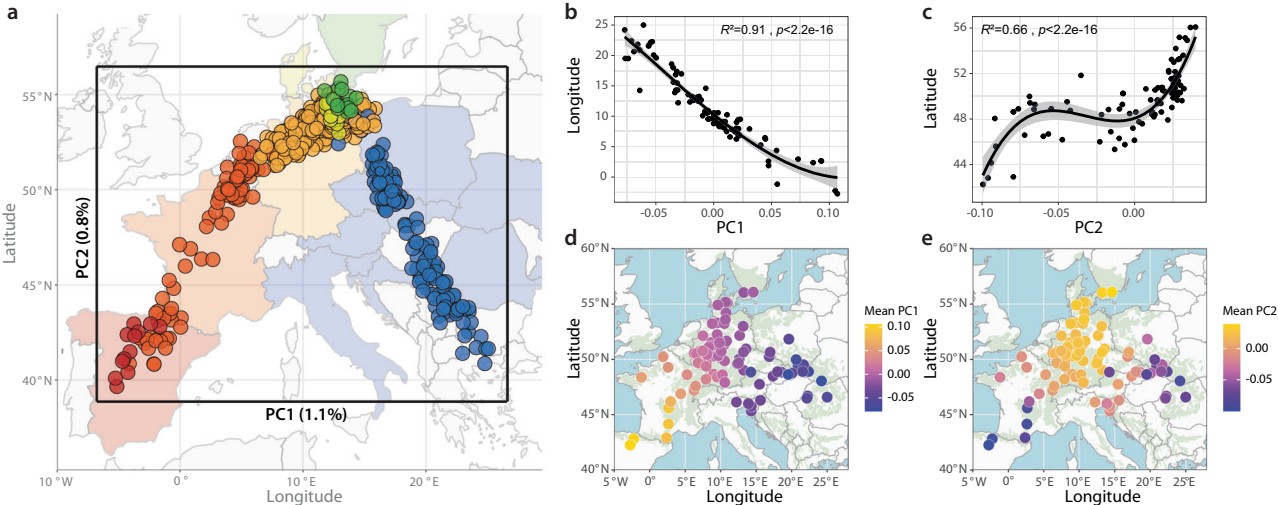

**Fig. 1 | Genomic variation across the distribution range of European beech (*Fagus sylvatica*) mirrors geography. a** Overlay of a map of Europe and the results of a principal component analysis (PCA) using 540k independent genome-wide variants in 653 largely unrelated (less than 2nd degree) individuals from 98 populations shows correlation between geography and PC1 and PC2, which explain 1.1% and 0.8% of the total genetic variation, respectively. Each point represents one of the 653 individuals plotted according to their PC1 and PC2 values (Source Data 1). Colors correspond to the country of the origin of their source populations: Spain = red,

France = orange, Germany = yellow, Denmark = light green, Sweden = dark green, other countries = blue. **b, c** Polynomial regression (3rd order) of average PC1 and longitude (b), and PC2 and latitude (c) reveals highly significant ($p < 2.2e\text{-}16$) relationships (adjusted $R^2$ = 0.91 and 0.66, respectively). Each population is marked by a black point, bold lines and shading indicate regression and 95% confidence interval of the model. **d, e** Geographic origins (longitude and latitude) of the 98 analyzed populations are depicted by circles. Colors indicate the population means of PC1 (**d**) and PC2 (**e**). Green background shading represents the distribution range of beech.

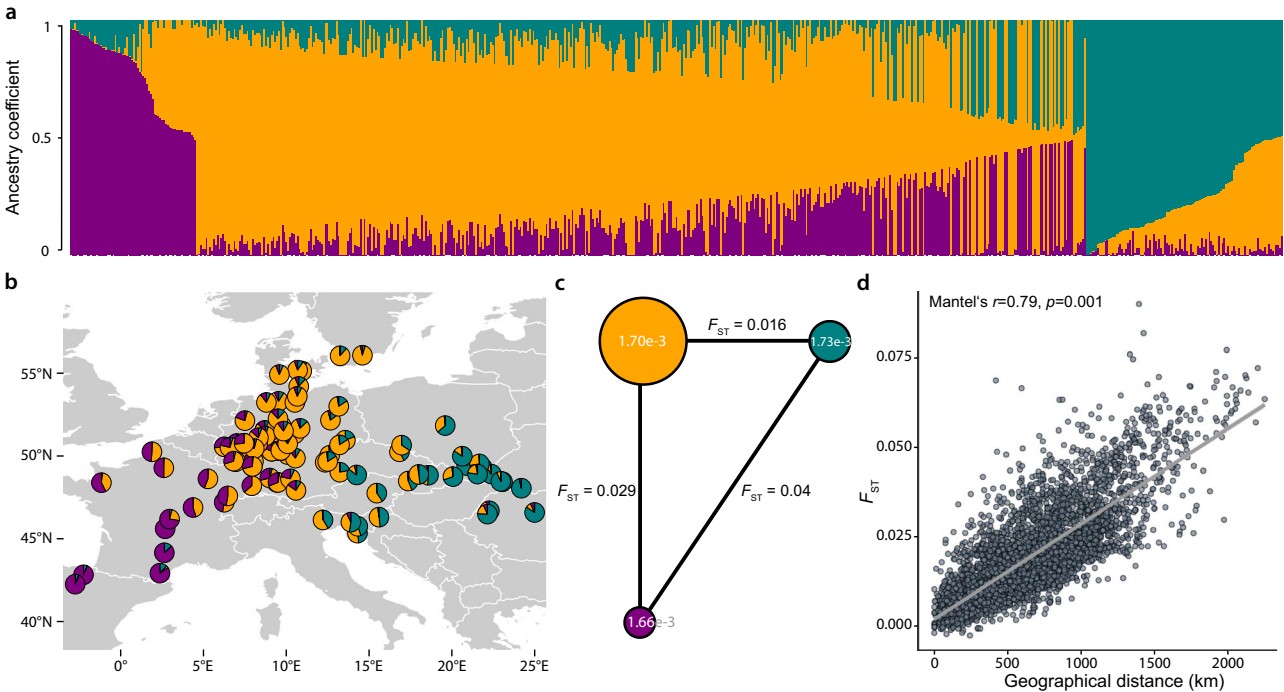

**Fig. 2 | Population genomics reveals three ancestral genetic clusters and significant isolation-by-distance. a** Analysis of the individuals' ancestry coefficients indicated the presence of three major ancestral genetic clusters (K = 3) whose proportions are represented for each of the 653 unrelated beech individuals by purple, orange and green bars, sorted by cluster membership (Source Data 2). **b** Most populations belong to the central cluster shown in orange, while the clusters shown in green and purple dominate in the East and in the West, respectively.

**c** Genetic diversity (π) indicated by white numbers within the circles is similar among the three clusters, which are depicted as circles scaled by the number of individuals with more than 70% cluster membership. Genetic differentiation ($F_{ST}$) shown by the length and the numbers on the lines increases with geographic distance. **d** Pairwise genetic differentiation ($F_{ST}$) between populations (Source Data 3) compared to geographic distances reveals significant (Mantel's $r$ = 0.79, $p$ = 0.001) isolation-by-distance. Gray line depicts corresponding linear model.

## Genotype-environment association (GEA) analyses suffer from pervasive random signal

To explore the genetic basis of local adaptation, we extracted the 19 bioclimatic variables from the WorldClim database[31]. These

interpolated climatic indices provide a powerful resource to assess GEAs, although other environmental data types and sources may yield additional insights[13]. As expected from the geographic origins of our populations, the bioclimatic variables show broad mostly normal

distributions (Supplementary Fig. 6). Non-clinal genetic patterns arising from natural selection under certain scenarios can limit the fraction of locally adaptive sequence variants that can be accurately inferred[32]. Nevertheless, well-powered landscape genomics datasets may still enable the detection of adaptive clinal variation. In line with this, our data exhibited thousands of statistically significant (Bonferroni correction, $p < 9.25e\text{-}8$) GEAs scattered across the genome in a latent factor mixed model (LFMM) analysis[33]. However, performing the same analysis with randomized geographical coordinates revealed a highly similar signal of statistically significant associations, even after false discovery rate (FDR) or Bonferroni correction. Since the genotypes are not associated with the random coordinates in any biologically meaningful way, this signal is non-causal by definition (Supplementary Fig. 7). Importantly, this indicates that running a genotype-environment association analysis and simply correcting for multiple testing may yield an exceedingly high number of false positives, that is variants that are statistically associated but have no biological relevance.

High levels of false positives, of close to 100%, have already been reported in simulation studies for different GEA methods[32,34]. To identify the genetic basis of local adaptation, true signal thus needs to be separated from the apparently overwhelming signal of random associations. We aimed to tackle this challenge by comparing the $p$ value distribution of random GEA runs with that of our real data to estimate the likelihood of a signal to occur by chance (Supplementary Fig. 7). While this, as any hard threshold, cannot differentiate causal from non-causal associations and will thus inevitably lead to false negatives (i.e., true associations not being identified), it provides a measure of confidence for the potential adaptive variants in a specific dataset. Using this randomization approach to determine a significance threshold for which the likelihood of being exceeded with random data compared to the real data is 5% (number of associations with random data/number of associations with real data = 1/20), only one single locus on chromosome 2, associated with winter temperature, remained (Fig. 3a, Supplementary Fig. 8). Lowering the threshold to 50%, a level at which half of the associations are expected to occur by chance, eight additional loci were identified (Fig. 3a).

## A highly associated locus for winter cold may cause adaptive differentiation by modulating phenology

The single high-confidence locus on chromosome 2 also stands out when using all 3.68 million variants (Supplementary Fig. 9) or other GEA methods, such as BayPass[35] or WZA[34] (Fig. 3b–d). However, with random data these methods pick up just as many associations as with the real data (Supplementary Fig. 10). The same is true for the multi-variate RDA method[36] which shows that a substantial part of the explainable genetic variance in our data can be attributed to differences in the environment (Supplementary Fig. 11, Supplementary Table 1). Isolation-by-environment (IBE) of associated variants, corrected for geographic structure, sometimes used as an indicator of local adaptation, was significant (partial Mantel test, $p < 0.001$) for real and random data thus also not serving to distinguish causal from non-causal signal (Supplementary Fig. 12). A more detailed view of the high-confidence genomic region shows extended linkage disequilibrium (LD) across approximately 300 kb (Fig. 3e). Thus, a natural sequence variant within this region is likely involved in the adaptation to different winter temperatures.

The projection of the most significantly associated variant onto the map shows higher frequencies of the alternative haplotype, associated with lower winter temperatures, in the eastern part of the range but also in some high-altitude populations such as provenance 11 from central France (Supplementary Figs. 1 and 3f). At the same time, the map of haplotypes highlights that the associated sequence variants represent common alleles that are not fixed and confined to specific populations or geographic regions but are segregating across the

entire distribution range. Notably, the reference allele of the cold-associated haplotype dominantly delays spring phenology by about 2.5 days (one-way ANOVA and Tukey's *post hoc* HSD test, $p < 0.001$, Fig. 3g) which could contribute to frost avoidance in the less predictable climates of the Atlantic part of the range. Interestingly, one annotated gene in the region, encoding a Callose synthase 1 (Bhaga_2.g94), has been shown to play an important role in controlling winter dormancy in European aspen (*Populus tremula* L.)[37]. This gene was highly expressed, albeit not differentially expressed between genotypes, in winter buds of eight beech trees sampled in December (Fig. 3h). It could thus differentially regulate winter dormancy and spring bud burst as an adaptation to different climates.

## Common garden and reciprocal transplant experiments suggest extensive phenotypic plasticity

The winter cold locus also shows up in a local analysis, which can increase statistical power[38], using individuals from the central genetic cluster only (Supplementary Figs. 13 and 14). Given its strong association across different geographic scales and its effect on potentially adaptive phenotypic differentiation, one might expect it to have an effect on fitness. Trees carrying the locally adaptive reference allele should show higher levels of fitness in our common garden in northern Germany than trees with the alternative allele. In beech, one important proxy for fitness appears to be growth with larger trees generating more offspring[39]. Surprisingly though, allelic variation of the winter cold locus did not affect growth, that is stem circumference, of the beech trees in our common garden in northern Germany (one-way ANOVA, $p = 0.51$, Supplementary Fig. 15). One explanation for this could be conditional neutrality, where the locus is beneficial only in one but neutral in the other climate. To test this possibility, we performed a reciprocal transplant experiment by analyzing two common gardens strongly differing for the minimum temperature of the coldest month and comprising provenances from the two contrasting local climates (Fig. 4a). Strikingly, however, again we did not find any indications of local adaptation when analyzing growth, i.e., stem diameter at breast height (DBH), or survival at age 30 (Fig. 4b, c). While we identify provenance- and site-specific differences for growth (but not survival), there is no difference between groups of local provenances nor a significant genotype-environment interaction (two-way ANOVA, $p = 0.42$ and $p = 0.51$, respectively, Fig. 4b, c) which is considered a hallmark of local adaptation[40]. Given the consistent genetic differentiation between these local populations (Supplementary Fig. 16) and the large climate distance between the two common gardens, our results suggest extensive phenotypic plasticity.

Does this phenotypic plasticity in our common gardens thus imply the absence of local adaptation in beech? And does it mean that any beech population could grow equally well anywhere across the range? While this can be true for a single planted generation under specific environments, the situation may markedly change under different environmental conditions, and especially in natural forest ecosystems. With competition by other species, rare extreme weather events and natural selection during the seedling stage, relevant performance and fitness differences that are masked in our artificial common garden system could become visible. The signal identified in our genotype-environment association analyses supports the importance of adaptive phenotypic differentiation in natural beech ecosystems. We therefore wanted to further explore this differentiation beyond the single high-confidence locus identified in our GEA analysis by characterizing the genetic architecture of spring phenology in a genome-wide association study (GWAS) for bud burst.

## Genome-wide association studies (GWAS) reveal polygenicity and missing heritability

Phenology is geographically differentiated across the distribution range of beech[41] and has been indicated to be of adaptive relevance in various tree species[42,43]. In poplar, a single major locus has been shown

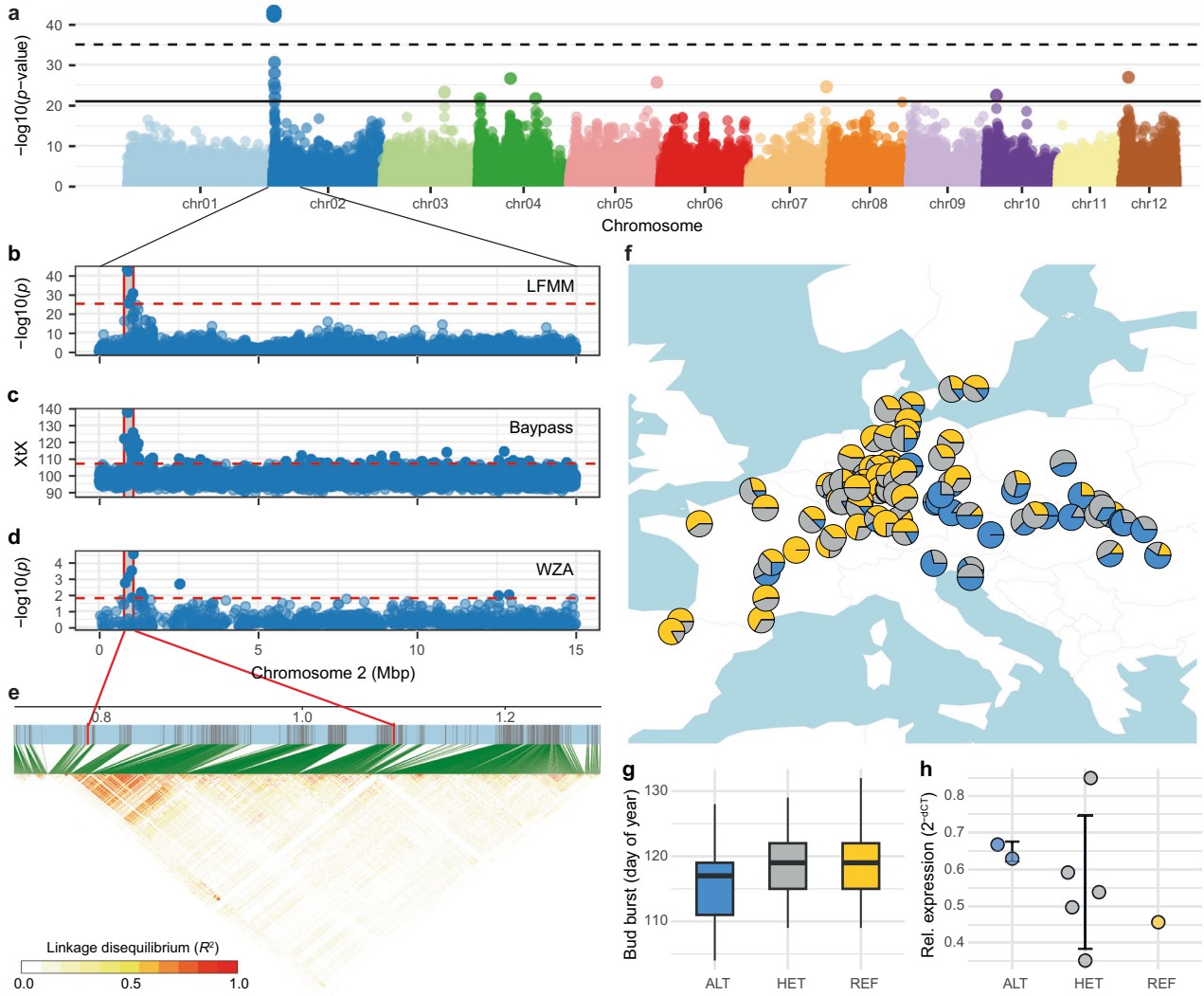

**Fig. 3 | Genotype-environment associations reveal genomic signals of local adaptation. a** Manhattan plot of the 540k LD-filtered genome-wide variants in 653 individuals from 98 populations. The significance of the association of each variant along the 12 beech chromosomes with the minimum temperature of the coldest month (bio6) of the population origins is indicated by the -log10 $p$-values on the y-axis (Source Data 4). The dashed and solid horizontal lines indicate significance thresholds for which the likelihood with random data compared to the real data is 5% and 50%, respectively (Supplementary Fig. 7). **b, c, d** Close-up view of the LFMM results (**b**) and two alternative methods, BayPass XtX (**c**) and WZA z-scores (**d**), of the highest confidence genomic region on chromosome 2 delineates the locus to a 300 kb region (0.79–1.09 Mbp). **e** The pairwise linkage disequilibrium ($R^2$) for non-LD-filtered variants is shown. Red colors indicate high $R^2$ values. The associated genomic region is marked by red lines. Green lines connect LD values to physical positions on the chromosome. **f** Pie charts show geographic distribution of genotypes for the top LFMM hit (chr2:898,040); colors indicate genotypes homozygous for the reference allele (REF, yellow), heterozygous (HET, gray) or homozygous for the alternative allele (ALT, blue). Only populations with more than three individuals are depicted. **g** Boxplots show spring phenology, i.e. day of bud burst, for the unrelated beech individuals in the common garden ordered by their genotype for the same variant shown in (**f**) (ALT: $n = 157$, HET: $n = 265$, REF: $n = 229$). Boxes indicate the interquartile range with the whiskers extending this range no more than 1.5 times and the center depicting the median. The REF allele dominantly causes later bud burst (one-way ANOVA and Tukey's *post hoc* HSD test, $p < 0.001$). **h** Relative expression ($2^{-dCT}$) of the *CALLOSE SYNTHASE 1* gene in winter buds was determined for eight trees using qRT-PCR. Circles represent relative expression, error bars ± 1 standard deviation from the mean.

to control local adaptation for autumn phenology, i.e. the timing of bud set[44]. A single locus affecting the timing of spring phenology also turned up in our GEA analysis. Does this indicate a simple genetic architecture with a single major locus driving phenotypic differentiation in phenology in beech as well? To address this question, we performed a GWAS for bud burst assessed during two years in the same 653 beech individuals as used in the GEA analysis. While we can predict phenotypic variation for bud burst with an accuracy of up to 0.45 using genomic prediction models (Supplementary Fig. 17) we did not identify any consistent GWAS loci (Supplementary Fig. 18). Significant associations changed between years, despite a high correlation of phenotypes (Pearson's $r = 0.889$, $p < 2.2e-16$), and only explained a small fraction of the overall phenotypic variation (1.4–5.1%,

Supplementary Fig. 18) demonstrating high levels of missing heritability. These results indicate a complex genetic architecture with a large number of small-effect loci resulting in insufficient statistical power for genetic dissection. Growth, which as mentioned before is a proxy for fitness in beech[39], is genetically even more intractable, with virtually no explanatory power of genomic prediction and no significant associations in our GWAS (Supplementary Figs. 17 and 18).

Importantly, GEA analyses are conceptually similar to GWAS, only with unmeasured composite fitness phenotypes approximated by the environment instead of direct phenotypic measurements. Thus, it may not be surprising that the genetic basis of environmental adaptation cannot be resolved with relatively small sample sizes of less than 1000 individuals either and that heritability remains largely

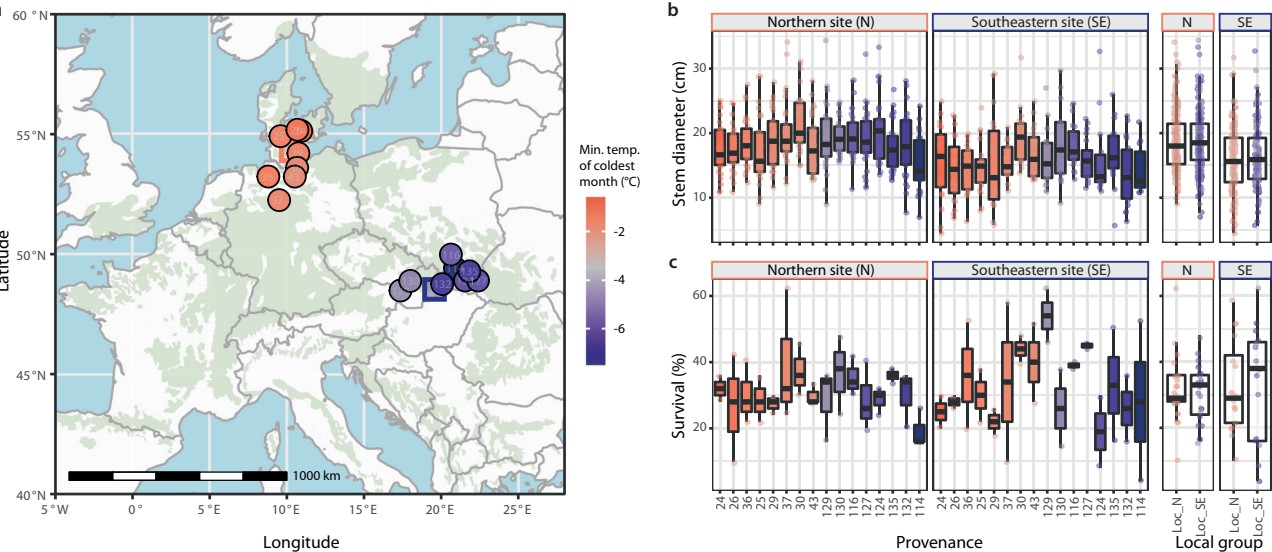

**Fig. 4 | Reciprocal transplant experiment suggests extensive phenotypic plasticity. a** We selected two common gardens, indicated by rectangles, strongly differing for the minimum temperature of the coldest month (bio6) and comprising two contrasting sets of local provenances, eight from the Atlantic and eight from the continental climatic region, depicted by circles. Colors indicate average minimum temperatures, gray marking the median value across the beech distribution range, which is shown by green shading. **b, c** Boxplots show growth, i.e., stem diameter at breast height (DBH), for the best 10 out of 50 originally planted trees in each of three blocks at the Northern (N) site ($n = 30$) and two blocks at the Southeastern (SE) site ($n = 20$) (**b**) (Source Data 5) and survival among these 50 trees for each of the blocks ($n = 3$ and $n = 2$ in the N and the SE site, respectively) (**c**) (Source Data 6), for eight Atlantic and eight continental provenances, individually and combined. Boxes indicate the interquartile range with the whiskers extending this range no more than 1.5 times and the center depicting the median. Additionally, all individual values are plotted as points. Colors of boxes and points follow the same color scheme as in (**a**).

missing[45–47]. Taken together, we interpret our single high-confidence GEA locus as an indication of high missing heritability caused by a polygenic genetic architecture. We find additional indications of this polygenic signal in our GEA results. For some of the bioclimatic variables, e.g., isothermality (bio3), the ratio of real vs. random signal increases relatively steeply already for lower-significance variants (Supplementary Fig. 19), suggesting the presence of many small-effect loci below the formal significance threshold. The machine learning algorithm "Gradient Forests," which can be used to model turnover functions of allele frequencies in response to environmental variables[48], also highlighted bio3 as the most important bioclimatic variable in our dataset (Supplementary Fig. 20) even though no individual locus stood out (Supplementary Fig. 8). Just as using our genomic data for reliable prediction of bud burst despite not having the power to dissect the individual loci in a GWAS, we might be able to predict maladaptation to future environments using 'genomic offset' models without knowing the underlying genetic basis.

### Genomic offset reveals broad- and fine-scale patterns of potentially mal-adaptive genetic variation

Even if genomic offset between current and future conditions, based on current genotype-environment associations, may not always predict future performance due to compensating effects of plastic responses or differences between local adaptation (local vs. foreign) and future maladaptation (home vs. away)[21], it can still reveal interesting spatial patterns of potentially adaptive genetic variation in relation to climate change. We therefore assessed genomic offset in our data by employing two conceptually different methods. First, we calculated 'risk of non-adaptedness' (RONA) for isothermality (bio3), which as described above exhibits the strongest signal of polygenic adaptation in our data, to estimate the distance between current and putatively required future allele frequencies weighted by the strength of their linear associations[49,50]. Second, we used the Gradient Forests (GF) method, to calculate the genomic offset using nonlinear turnover functions and all 19 bioclimatic variables[48].

The two methods, which were significantly correlated (Pearson's $r = 0.3$, $p < 0.01$, Supplementary Fig. 21), both indicated higher offsets in the southwestern part of the range (Fig. 5). Due to the extensive phenotypic plasticity in our common gardens only a small fraction of the phenotypic variance for growth is explained by the populations' origins (one-way ANOVA, adjusted $R^2 = 1.8\%$, Supplementary 21). We could therefore not validate our genomic offset estimates, even though RONA values calculated with respect to our common garden instead of future climate in a space-for-time approach exhibited a correlation with stem circumference (one-way ANOVA, adjusted $R^2 = 0.9\%$, $p < 0.01$, Supplementary 21). One possibility to validate our genomic offset estimates in the future and to evaluate the impact of different environmental variables may be the assessment of our populations in their present natural habitats. Especially after recent years with extreme weather conditions such as heat and drought, possible differences in current maladaptation should become apparent.

For a more holistic view of species' responses to climate change, it was suggested to consider additional variables such as gene flow, dispersal or genetic load[8]. Recessive genetic load was independent of genomic offset in our data highlighting another layer of complexity (Supplementary Fig. 21). It also did not follow any particular geographical pattern suggesting that demographic history in the sampling range did not involve extreme range expansions or colonization events affecting effective population size (Supplementary Fig. 22), in agreement with the similar levels of nucleotide diversity found across genetic clusters. Most importantly, our genomic offset analyses highlighted geographically fine-scale variation (Fig. 5). This result is consistent across different climate change models and scenarios (Supplementary Fig. 23). It is nicely exemplified by some populations from southern Germany, where adjacent populations exhibit markedly different levels of genomic offset (Fig. 5, Supplementary Fig. 24). This pattern appears to be only partially explained by the distribution of predicted climate change (Supplementary Fig. 24) indicating the presence of fine-scale differences in potentially mal-adaptive genetic

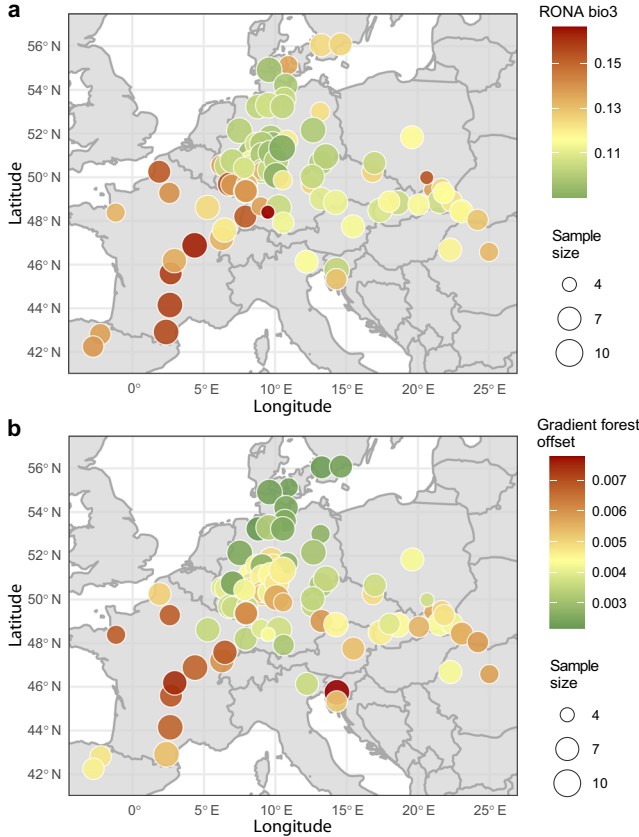

**Fig. 5 | Genomic offset reveals broad- and fine-scale patterns of potentially maladaptive genetic variation across the landscape. a, b** The 'risk of non-adaptedness' (RONA) for isothermality (bio3) (**a**) (Source Data 7) and gradient forests offset for all 19 bioclimatic variables (**b**) (Source Data 8) were modeled using 3321 associated (Bonferroni-adjusted $p < 0.05$) and 10,000 randomly selected variants, respectively. Colors show estimated offset values, with yellow indicating the median. The size of the circles marks the number of individuals per population ($n = 4$–10). Predicted climate in 2081–2100 for an intermediate climate change scenario (RCP4.5, MPI-ESM1-2-HR) was compared to near-current (1970–2000) conditions.

variation between some populations and highlighting the importance of spatial resolution.

## Discussion

The genomic analysis of 100 beech populations across the distribution range revealed a striking correlation between genetics and geography. The autochthonous populations may thus be locally adapted to different environmental conditions. Despite high levels of phenotypic plasticity under experimental common garden settings, we identified genomic signals of local adaptation. Importantly, our analyses emphasize the importance of data randomization and the need for larger sample sizes to achieve sufficient statistical power given the expected polygenic nature of adaptive traits. It will be exciting to see collaborative landscape genomics studies with a much higher number of sequenced trees. Only then can we start to fully appreciate the polygenic signal of adaptive differentiation, which will be important for further developing prediction models of forest growth and stability under climate change.

Our GEA and GWAS analyses suggest high levels of (potentially adaptive) standing genetic variation, which might contribute to adaptability. However, before any practical implementation, empirical validation of the relevance of this polygenic signal will be critical. Smartly designed new common garden experiments could contribute

to this[21]. There should be an explicit focus on the specific effects of environmental variables. Under different environmental conditions, different environmental variables and thus different adaptive alleles will be most important for shaping the species adaptive responses, especially in the case of conditional neutrality. Additionally, future experiments should focus on more stressful sites similar to those expected under climate change. This might unmask so far hidden natural variation between populations. The possible relevance of rare extreme weather events, competition and natural selection during the seedling stage should be considered for the experimental design. Also, different types of experiments may be needed depending on whether the research focus is silvicultural or ecological. Finally, for a reliable view of the adaptive genomic variation, spatial resolution is critical as a continuous distribution of adaptive alleles and interpolation may not be warranted[20]. Given that the selection of loci analyzed apparently does not play much of a role[15], this should become increasingly feasible using moderate density genotyping platforms. Large genetic biodiversity monitoring projects are being discussed and high-resolution sampling of different plant and animal species are on the way[51,52].

In conclusion, using genomics to predict future species performance and ecosystem stability is still a rapidly developing field. Especially in long-lived trees it promises great potential for informing forest management and conservation[53]. Careful variable selection and rigorous validation will be necessary before practical implementation. Our work using extensive genome-wide data across the range of a foundation forest tree species shows the importance but also the complexity of using genomics to resolve patterns of local adaptation and better understand climate change effects. Due to limited sample size and the polygenic architecture of adaptive traits, we are only skimming the surface of the encoded genomic information. Additionally, our common garden data point to the possibility of plasticity overlaying the expression of adaptive differentiation. Nevertheless, with increasing genomic and phenotypic data and new experimental sites, enhanced prediction models will likely contribute to conserving and restoring forests for biodiversity and as one of the most important carbon sinks for climate change mitigation.

## Methods
### Plant material and sampling
For the sample collection of beech genotypes from across the distribution range, we employed one of 23 common garden experimental sites of an international beech provenance trial planted in 1995. Specifically, we used the site near Schädtbek, Germany (trial code BU1901), where individuals from 100 different populations are growing in a randomized complete block design with multiple-tree plots of originally 50 individuals[54,55]. The coordinates of these populations (provenances) and the trial site are given in Supplementary Data 1. In each of the three blocks, three individuals were randomly selected from each population, totaling 900 samples. The only selection criterion was that the crown of the sampled trees is part of the canopy, to allow phenotyping with unmanned aerial vehicles (UAVs). Tree circumference at breast height was measured for all sequenced trees of each of the 100 populations in December 2022. Spring phenology was determined every two to three days from mid of April until mid of May 2022 and 2023 and defined as the majority of the tree crown showing leaves emerging from the winter buds. The phenotypic measurements are provided in Supplementary Data 4.

### Reciprocal transplant experiment
Our study site in Schädtbek (BU1901) is part of the 4th international beech trial series sown in 1993 and planted in 1995[54]. A parallel site with a comparably large number of provenances is Vrchdobroč (BU1905) in Slovakia. Located at 840 m above sea level in the continental part of the distribution range, it experiences a contrasting climate with much lower minimum temperatures. This site was planted with two

replications. During dormancy in 2022/23 a planned full assessment of 30-year diameter at breast height (DBH) was conducted at both sites. Here, we focused on eight provenances of local proximity to each trial site, 16 in total (northern provenances: 24, 25, 26, 29, 30, 36, 37 and 43; southeastern provenances: 114, 116, 124, 127, 129, 130, 132 and 135) to test for an effect of local adaptation. Analysis of variance (ANOVA) was conducted based on the following linear mixed model: $Y_{ijk} = \mu + P_i + E_j + (P \times E)_{ij} + B(E)_{jk} + e_{ijk}$, where $Y_{ijk}$ is the phenotypic observation of a trait made for the $i$th provenance ($P$), grown at the $j$th environment ($E$), located in the $k$th block ($B$) within environment $E$. $P \times E$ represents the provenance by environment interaction, $\mu$ is the overall experimental mean, and $e$ is the experimental error (residual). Further, the model was applied with two provenance groups rather than individual provenances, each aggregating one set of eight local provenances. Response variables were DBH of the 10 strongest, most competitive trees per plot (Source Data 5), and survival percent per plot derived from the number of living trees relative to their number at establishment (Source Data 6).

### DNA extraction and Illumina resequencing
Young leaves were sampled in the field in May and June 2021 and placed on ice in collection microtubes (QIAGEN, Hilden, Germany) in a 96-well format. Samples were stored at −20 °C until DNA extraction, following a previously described protocol[56]. DNA sample QC, library preparation and sequencing were performed by Novogene Europe (Cambridge, UK). Of the 900 samples, 26 failed sample QC, probably due to low amounts of starting material. For the remaining 874 samples, Illumina sequencing libraries were prepared and an average of 22.98 Gb of 150 bp paired-end data, which corresponds to approximately 42.5x coverage of the beech reference genome[29], were generated on the Illumina Novaseq 6000 platform with an S4 flow cell. Additional details on the sequencing statistics can be found in Supplementary Data 2.

### Variant calling
Variant calling was performed using GATK version 4.0.5.1 following the best practices for germline short variant discovery where possible[57]. After read filtering, which consisted of (1) removing reads containing adapters (5′ Adapter: 5′-AGATCGGAAGAGCGTCGTGTAGGGAAAGAGTGTAGAT CTCGGTGGTCGCCGTATCATT-3′, 3′ Adapter: 5′-GATCGGAAGAGCACA CGTCTGAACTCCAGTCACGGATGACTATCTCGTATGCCGTCTTCTGCTT G-3′), (2) removing reads containing more than 10% undetermined bases, and (3) removing reads of low quality, that is reads with a Qscore < = 5 for more than 50% of the bases, the clean data were mapped against the chromosome-level beech reference genome[29] using bwa-mem version 0.7.12 (Ref. 58) with the following parameters: -k 32 -M -R. Duplicate reads were marked with Picard tool's (v2.26.2) 'MarkDuplicates' (http://broadinstitute.github.io/picard/). Using GATK's 'HaplotypeCaller', gVCF files were generated which were combined by GATK's 'CombineGVCFs'. Finally, joint genotyping was performed using GATK's 'GenotypeGVCFs' (Ref. 59).

### Variant filtering
Hard variant filtering, instead of the recommended VQSR that requires a validated set of variants as a truth or training set not available for beech, was based on variant quality scores, variant coverage, missing data, linkage disequilibrium, minor allele frequencies and heterozygosity. Specifically, we first filtered the SNP and indel data independently, considering the recommendations from the technical documentation on "Hard-filtering germline short variants" on the GATK website. We therefore removed variants based on strand bias (FisherStrand (FS) > 60 & StrandOddsRatio (SOR) > 3) and mapping quality (RMSMappingQuality (MQ) < 40, MappingQualityRankSumTest (MQRankSum) < −12.5). Based on the distribution of the variant confidence score QualByDepth (QD) we choose a more stringent cutoff of QD > 10, to remove any low-confidence variants. Filtering was performed with bcftools v1.7 (Ref. 60).

We then extracted the variant sequencing depth values (DP) using vcftools v0.1.16 (Ref. 61) to visualize the DP distributions. To avoid any potentially hemizygous or duplicated sequence variants not resolved in the reference genome, we chose relatively stringent coverage cutoffs based on the mode of the distribution. We only allowed for DP values from 25% below to 50% above the mode of the distribution, which represent values in between haploid and diploid or diploid and tetraploid coverage, respectively. This resulted in cutoffs from 24.9 to 49.8 for SNPs and from 24.8 to 49.7 for indels. Additionally, we filtered out variants with more than 10% missing data and only kept biallelic variants. This resulted in a final dataset of 11.97 million variants for 874 individuals.

Using PLINK v1.9 (Ref. 62) we first checked for general patterns in our dataset using a principle component analysis (--pca). This analysis highlighted nine individuals, all from the easternmost provenance from Bulgaria, as outliers (Supplementary Fig. 3). Since the origin of the Bulgarian provenance (provenance 158) is close to the distribution range of the second beech species in Europe, *Fagus orientalis* (https://www.euforgen.org/species/fagus-orientalis), these individuals may not represent pure *Fagus sylvatica* and were therefore removed from the dataset. Additionally, we excluded one population from Northern Germany that appeared admixed (provenance 32) and one individual from a German population (individual B9) that was closely related to individuals from a population from Slovakia (provenance 135) and may represent a planting error (Supplementary Fig. 3).

We then filtered for minor allele frequency of 0.01 and pruned variants in complete linkage disequilibrium (LD) in windows of 100 variants (--indep-pairwise 100 25 0.99). We also removed variants with extreme deviation from Hardy-Weinberg equilibrium ($p < 1e-08$)[63]. Finally, we checked for individuals with excessive levels of missing data or heterozygosity, as previously described in Refs. 64,65. We removed a single individual with high levels of missing data (78% vs less than 1.6% for all other individuals). Additionally, we removed eight individuals based on high levels of heterozygosity, exceeding three standard deviations from the mean.

To identify related individuals, we used the program KING v2.3.0 (Ref. 66) with 540 thousand mostly independent variants (LD < 0.2 in windows of 50 variants). This analysis identified 192 individuals with a relatedness of second degree (first cousins) or higher, which we removed using the function '--unrelated -degree 2' Our final dataset comprised 3.68 million variants, of which 540,566 variants exhibit LD values below 0.2 in windows of 50 variants, for 653 "unrelated" individuals.

### Population genomics
Employing the R package LEA v3.10.1 (Ref. 67), we imported our final 3.68 million total variants and 540,566 independent variants using the function 'ped2lfmm'. Using the independent variants, population genetic structure was evaluated with a principal component analysis and an admixture analysis that are both implemented in LEA and called by the pca()- and snmf()-functions, respectively. Both methods indicated three genetic clusters in our data. Thus, we determined ancestry coefficients with snmf using K = 3 (Source Data 2) and assigned our individuals into the three ancestral genetic clusters, determined by a 70% cluster membership probability threshold. Population differentiation (Weir and Cockerham's $F_{ST}$) between and nucleotide diversity ($\pi$) within these clusters were calculated in 10 kb non-overlapping sliding windows using vcftools v.0.1.16 (Ref. 61). It should be noted that $\pi$ generated using vcftools will have underestimated values due to the treatment of all missing sites as monomorphic. We then created an $F_{ST}$ matrix and the geographic distance (km) matrix between the populations for the isolation-by-distance (IBD) analysis. Mantel tests were run using the R package vegan v.2.4-6 (Ref. 68), with the significance being estimated based on 999 permutations.

## Genotype-environment association (GEA) analyses

For the analysis of genotype-environment associations we first extracted 19 bioclimatic variables representing historical climate data for the reference period from 1970–2000 from the WorldClim database[31] (v2.1) with a resolution of 5×5 km for all 98 populations using the R packages geodata[69] and terra[70]. This resolution approximately matches the precision of the geographic coordinates of our populations. As described above, we used the R package LEA v3.10.1 (Ref. [67]) to import the two final sets of variants (3.68 M and 540k) with the function "ped2lfmm." As our populations probably exhibit three main ancestral genetic clusters, we chose K = 3 latent factors to account for confounding effects caused by population structure. Missing genotype data were imputed using the impute()-function in LEA. We fitted the latent factor mixed model (LFMM) using the lfmm()-function in LEA with 5 repetitions, 10,000 iterations and a burnin of 5000. The bioclim variables were used as environmental file. *P*-values were extracted using the lfmm.*p*values()-function and a conservative Bonferroni significance threshold was applied.

Additionally, we performed three randomizations of the geographic coordinates of the 98 populations to randomize the environmental variables but maintain population structure. We used those random environmental data to re-run the LFMM analysis (with 3 repetitions, 6000 iterations and a burn in of 3000). *P*-value distributions of the three random runs were compared with the *p*-value distribution from the run using the real data to determine a significance threshold for which the likelihood of being exceeded with random data compared to the real data is 5% (number of associations with random data/number of associations with real data = 1/20) or 50% (Supplementary Fig. 7). Finally, we ran a "local" LFMM analysis with individuals from the central genetic cluster only. For this, we selected populations with at least three individuals with more than 70% membership to the central genetic cluster (Supplementary Fig. 13, orange). Again, we also ran the local LFMM analysis with randomized environmental data.

Further, we applied BayPass v2.4 (Ref. [35]) and WZA (Ref. [34]) as two additional methods for calculating GEAs. Within BayPass we performed three replicate runs of the core model to estimate XtX and population covariance matrix and then took the median values across independent runs. Using the median covariate matrix, we ran three repetitions of the auxiliary covariate model and kept the median values of Bayes Factor (BF) across replicate runs. To get the threshold values we constructed null distributions of the XtX values, by performing simulations of pseudo-observed data (POD) using the simulate.baypass() function. Significant threshold for BF was 10 (i.e., Jeffrey's rule for "strong evidence"). For WZA we calculated allele frequencies using vcftools v.0.1.16 freq2 option[61]. Population allele frequencies were correlated with 19 bioclimatic variables using Kendall's τ statistic in R. Genome-wide τ results were analyzed in 10 kb non-overlapping windows using the weighted-Z analysis (WZA)[34]. Both BayPass and WZA analysis were repeated with randomized population coordinates and thus randomized environments for the bio6 environmental variable.

We additionally applied the multivariate method redundancy analysis (RDA)[36] to calculate GEAs. This analysis was performed using the R package vegan v.2.4-6 (Ref. [68]). To avoid overfitting, we selected 12 variables, removing highly correlated ones (such as "maximum temperature of warmest month" and "mean temperature of warmest quarter", or "precipitation of wettest month" and "precipitation of wettest quarter") keeping the more specific variables, i.e., daily or monthly rather than quarterly data. For further analysis, these variables were standardized (i.e., subtracted the mean and divided by standard deviation). We first applied partial RDA (pRDA) to dissect the effects of climate, population structure, and geography using four models (Supplementary Table 1). We used 12 bioclimatic variables representing climate ('clim' in our model), population scores along the first six PCA axes performed on 540,566 variants as a proxy for population structure ('struct' in our model) and longitude and latitude to represent geography ('geog' in our model). Population allele frequencies were used as a response variable in all models. We then used pRDA to identify candidate loci associated with the environment while correcting for population structure. We defined outliers based on the previously described rdadapt function[71]. The same analysis was repeated for three sets of randomized population coordinates and, thus, randomized environments.

Calculation of $R^2$ and visualization of LD pattern was done using LDBlockShow v.1.40 (Ref. [72]). For expression analysis of the Callose synthase 1 gene (Bhaga_2.g94), winter buds of eight beech trees were collected in December 2022. Total RNA was extracted with the Spectrum Plant Total RNA kit (Sigma-Aldrich, USA) according to the manufacturer's manual, Protocol A. Following that, DNase I digestion was performed using the Turbo DNA-free kit (Invitrogen, USA). For cDNA synthesis, 2 μg of RNA, Oligo (dT) primers and SuperScript IV reverse transcriptase (Invitrogen, USA) were used following the manufacturer's protocol, using 10 μl reactions without RNaseOUT. Reverse transcriptase quantitative PCR (qRT-PCR) was carried out in triplicates on a CFX96 Touch Real Time PCR Detection System (Bio-Rad Laboratories GmbH, USA) using the SsoAdvanced Universal SYBR Green Supermix (Bio-Rad Industries, Inc., USA) and a two-step PCR program with annealing temperature of 60 °C. Relative expression levels were calculated using the $2^{-\Delta Ct}$ method[73]. The primers are given in Supplementary Data 5.

## Genomic prediction and genome-wide association studies (GWAS)

For genomic prediction, we used the kin.blup() function from the R package rrBLUP v4.6.3 (Ref. [74]). For computing the additive relationship matrix with the A.mat()-function, we randomly selected 30,000 sequence variants from the final set of 540k LD-filtered variants. For each size of training population (*n* = 150, 250, 350, 450 and 550) we ran 200 cross validations by randomly assigning the according number of individuals, from the final set of 653 unrelated individuals, to the training and the validation populations. Predictive accuracies were calculated using the Pearson's correlation between predicted and measured phenotypic values. Phenotypic values included bud burst determined in two years and stem circumference measured in one year as described above and given in Supplementary Data 4. To identify genotype-phenotype associations, we performed GWAS using the R package GAPIT v.3.1.0 (Ref. [75]) using the BLINK model[76]. As a phenotypic input, we again used bud burst assessed during two years (2022 and 2023) and stem circumference measured in one year in the same final 653 individuals. A Bonferroni multiple test threshold was used to determine significance.

## Isolation-by-environment (IBE)

Using vcftools v.0.1.16 (Ref. [61]), we calculated pairwise $F_{ST}$ between 98 populations separately for significant (*q* < 0.01) and non-significant variants (*q* > =0.01), as identified in LFMM analyses with real and random data. We generated the $F_{ST}$, the geographic distance (km), and environmental distance matrices for both data sets. The environmental distance was calculated as Euclidean distance for the bio6 environmental variable. Partial Mantel tests were run using the R package vegan v.2.4-6 (Ref. [68]), with the significance being estimated based on 999 permutations. The same analysis was repeated with a randomized data set.

## Genetic load analyses

Population genetic load was evaluated by computing the average recessive and additive genetic load per individual, obtained by counting the average number of derived mutations with deleterious effects in homozygosis or total, respectively, relative to synonymous variants to adjust for heterozygosity differences across individuals (as in Ref. [77]). We first polarized variants using the *Q. robur* genome[78] and

retained 284,046 SNPs for which we were able to determine the ancestral state. Second, a custom database was constructed to perform genomic annotations (see below) and validated by comparing predicted protein and CDS sequences with those obtained from the Bhaga European beech reference genome[29]. Finally, SNP variants were annotated for their predicted effects on known genes (e.g., amino acid changes or loss of function mutations) using SnpEff[79] and used for the counts of deleterious and neutral-effect (synonymous) variants (based on 57,308 annotations).

## Genomic offset analyses

To estimate genomic offset of our studied populations, we employed the "risk of non-adaptedness" (RONA) and "Gradient Forests" offset measures[48,49]. Present (1970-2000) and future (2081–2100) bioclimatic variables for every population and for our common garden in Schädtbek (54°3' ''N 10°28' ''E) were extracted from the WorldClim database using the CMIP6 'MPI-ESM1-2-HR', the "BCC-CSM2-MR" and the "EC-Earth3-Veg" models[80–82] with a moderate (RCP 4.5) and a more pessimistic (RCP 8.5) shared socioeconomic pathway (SSCP) with a 2.5-minutes resolution (5×5 km). For RONA we used the bioclimatic variable isothermality (bio3). We calculated the population-specific allele frequencies of significant variants (Bonferroni-adjusted $p < 0.05$), as determined by the LFMM analysis, using the R package data.table[83] on the imputed genotype matrix as used for the LFMM analysis. RONA values were calculated as the distance between the current allele frequencies and the required frequencies under projected future climate conditions based on linear regressions (allele.freq-environmental.variable) weighted by the adjusted $R^2$ values of the linear models[50]. We randomly subsampled 10,000 SNPs for gradient forests (GF) analysis and removed six populations with less than five individuals. Using population allele frequencies and 19 bioclimatic variables, GF was conducted in R with 'gradientForest' function[84]. Following code previously described in Ref. 53, we calculated GF predictions that rescaled near-current and future bioclimatic variables into common units of genomic turnover. Genomic offset was then calculated as Euclidean distance between near-current and future genomic composition.

## Statistics and data visualization

For all statistical analyses and data visualization we used R v4.2.2 (Ref. 85) and the ggplot2 package[86]. To plot map data we used the R package rnaturalearth v1.0.1 (Ref. 87) or the software QGIS v3.32.0-Lima.

## Reporting summary

Further information on research design is available in the Nature Portfolio Reporting Summary linked to this article.

# Data availability

All sequencing data are available from NCBI's SRA under the accession number PRJNA1005581: https://www.ncbi.nlm.nih.gov/bioproject/PRJNA1005581. The vcf-files used for the analyses are available from EMBL-EBI's EVA under the accession number PRJEB80328: https://www.ebi.ac.uk/eva/?eva-study=PRJEB80328. Source data for all figures are additionally provided as Source Data files. Source data are provided with this paper.

# Code availability

Code used for the analyses is available under: https://doi.org/10.5281/zenodo.13305018.

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

## Acknowledgements

We would like to thank A. Eikhof, K. Groppe, S. Jencsik, S. Benischek and M. Spauszus for technical assistance in the lab and in the field. We are grateful to P. Eusemann and other members of the Thünen Institute of Forest Genetics, and K. Budde and other members of the AForGeN group for helpful comments and discussions. This work was funded by a core grant from the Thünen Institute and a grant from the FNR Forest Climate Fund (2219WK60A4). Support from the French government in the framework of the IdEx Bordeaux University 'Investments for the Future' program / GPR Bordeaux Plant Sciences is also acknowledged.

## Author contributions

Conceptualization: MF, BD, NAM. Data curation: DL, CG, KJL, DG, NAM. Formal analysis: DL, CG, KJL, MM, CB-J, I-LK, SG-M, NAM. Funding acquisition: BD, NAM. Investigation: DL, CG, NAM. Methodology: DL, CG, KJL, NAM. Supervision: DG, ML, SG-M MF, BD, NAM. Validation: CG, NAM. Visualization: DL, KJL, NAM. Writing – original draft: NAM. Writing – review & editing: All authors.

## Funding

## Competing interests

The authors declare no competing interests.
