## [Peer Review File · Nature Communications]

Genomic variation of European beech reveals signals of local adaptation despite high levels of phenotypic plasticityREVIEWER COMMENTS

Reviewer #1 (Remarks to the Author):

Dear editor, dear authors,

In this manuscript, the authors explore the genomic bases of local adaptation to climate in a European tree species – *Fagus sylvatica* – whole genome resequencing data for 865 individuals. After identifying a set of putative adaptive variants, they estimate a risk of maladaptation to climate change across the sampled populations. I believe the study to have great potential, but all the work is based on the assumed presence of local adaptation to climate in the studied populations of *Fagus sylvatica*, when the results of a large common garden and the present genomic results could suggest the opposite. If the authors do not provide evidence of local adaptation to climate in their sampling first, the study focus on genomic offset could become questionable.

My comments are listed below.

Global comments:

1) Genomic offsets are pertinent only if the species exhibit local adaptation to climate. The manuscript should convince the reader that it is the case for *F. sylvatica*. For example, what do the common garden results tell us? Has it been analyzed already in another study that the authors could refer the reader to? The last paragraph of the discussion seems to suggest that there are only very weak signs of local adaptation among the studied populations.

2) Assuming that the first point is resolved, the most interesting part of the study for me is the part where the authors explored the genomic bases of local adaptation to climate, including the polygenic nature of it, the potential link with negative selection, the key environmental drivers of adaptation, the genes identified, the consistency across different geographic scales... I would suggest shifting the focus of the manuscript towards these particular questions, that are still unanswered for *F. sylvatica* and even for a lot of tree species. That would mean conducting supplementary analyses (genetic variance partitioning in a multivariate way, GO terms enrichment, comparing results based on the geographic scale used...) but would make the study very interesting.

3) Showing the attempt of validating genomic offset predictions is necessary, even if it failed. However, to me, it could suggest here again that local adaptation to climate is too weak the studied populations of *F. sylvatica* to conduct this type of analyses.

4) Wording around the genomic offset metric in the introduction can sometimes be misleading: (Line 26) “estimates of species’ responses to climate change”; (Line 28) “account for dynamic population genetic processes”. Genomic offset does not predict a future intrinsic change in the population but only the amount of climate shift that will affect the current gene ~ climate relationship. Species or populations’ responses are not that easily predictable. This response, plastic change? decline? adaptation? extinction? migration? will depend on a lot of other factors that are not considered here. I believe it is critical information that must be carefully stated,

especially early on in the introduction.

Specific comments:

Line 26: Neither genomic offset nor SDM can actually “estimate species’ responses to climate change”. These metrics/models just predict where the genes ~ climate (genomic offset) or the presence ~ climate (sdm) relationship will be the most affect by a predicted change in climate. In other words, it estimates where species would have to respond the most to remain present or locally adapted, but never the response itself.

Lines 29-31: Integration of data or integration of concepts? These two last sentences of the paragraph are not very clear, consider revision.

Line 45: Consider splitting that sentence.

Line 53: Not sure what the authors mean when they say: “we set out to characterize...”. Maybe “we aimed at characterizing...” or “we sought to characterize...”?

Introduction: The Introduction mentions only very briefly some important concepts and ideas such as “maladaptation”; “genomic vulnerability”; “potential natural vegetation (PNG)”. I think it would be important to contextualize those a bit more.

Line 76: The authors state “remarkable correlation between the first two principal components (PCs) and geography” but only discuss PC1 in the following section, is PC2 correlated with latitude?

Line 78: What are the levels of genetic differentiation among populations? It should be mentioned here.

Line 80: I am not sure to understand how “these results demonstrate a high level of accuracy of the common garden”?

Figure 1: Panel b) and c) are redundant with panel a), and even potentially misleading because longitude is only relevant to PC1 (probably latitude to PC2?). The real underlying factor seems to be distance, as shown by supplementary figure 5, which would be much more appropriate in the main text, maybe as a panel of Figure 1. It is super interesting results.

Lines 84-85: What are the results sustaining the statement “Considering the large amplitude of environmental differences, relatively strong selection for adaptive differentiation can be expected”?

Lines 86-87: The link between IBD (the results of the paragraph) and local adaptation is not clear to me here. Do the authors mean that populations are exchanging genetic material and must then have reached their adaptive equilibrium? That adaptive allele should exhibit frequency clines?

Line 110-114: The link with negative selection is quite speculative but also potentially one of the most interesting findings of that study. I would recommend adding this result to the main text and exploring further this issue, for example backing it up with a real explanation of how purifying/negative selection could affect polygenicity and adaptive allele effect sizes in a

context of local adaptation.

Line 186-188: Because the results are only statistical predictions with no validation, I don't think the authors can say that their results "demonstrate the potential for employing putatively adaptive variation to predict general adaptive capacity to future climate conditions...".

Line 189-191: I don't see what in the results actually suggest that each forest stand should be assessed individually. How realistic would be to conduct a genomic study like this one on all important forest stands?

Line 193-195: How can the authors predict anything in their study if the link between genetic and environmental variation is too weak?

Lines 203-204: Genomic offset validation usually works fine even if the common garden is within the current climatic range of the species, see (Fitzpatrick et al., 2021; Capblancq & Forester, 2021; Lachmuth et al., 2023). Genomic offset is in fact very similar to the climate transfer distance approach, which has worked quite well for seed sourcing for a long time and for a lot of different tree species.

Line 186-213: The information given in the last paragraphs of the discussion are troubling, to me, they suggest that local adaptation is too weak in the studied populations to conduct such a study.

Lines 307-308: What threshold did the authors use to consider a locus as an outlier? Many "outliers" seem to still be in the overall genomic envelope in the Manhattan plots, making me wonder if the threshold value is appropriate. I would suggest comparing a False Discovery Rate approach with q-values to the Bonferroni correction. Just to make sure that many False Positives are not influencing the signal here. What about Q-Qplots and histograms of p-values provided by LFMM? These are usually very informative.

Thanks for the opportunity to review this manuscript. I am sure that the study could make an important contribution to the field once its foundations are strengthened.

Thibaut Capblancq

Literature cited:

Capblancq T, Forester BR. 2021. Redundancy analysis: A Swiss Army Knife for landscape genomics. *Methods in Ecology and Evolution* 2021: 2298–2309.

Fitzpatrick MC, Chhatre VE, Soolanayakanahally RY, Keller SR. 2021. Experimental support for genomic prediction of climate maladaptation using the machine learning approach Gradient Forests. *Molecular Ecology Resources*.

Lachmuth S, Capblancq T, Prakash A, Keller SR, Fitzpatrick MC. 2023. Novel genomic offset metrics integrate local adaptation into habitat suitability forecasts and inform assisted migration. *Ecological Monographs*: e1593.

Reviewer #2 (Remarks to the Author):

This study focused on the European key forest tree species, European beech (*Fagus sylvatica*), and reported a total of 653 whole-genome re-sequencing novel data, collected from 98 populations covering its entire distribution range. By employing population genetics analysis, environmental association analysis, and genetic offset prediction, the study assessed beech's landscape of future climate maladaptation risks and proposes recommendations for conservation strategies. Although the re-sequencing data used in this study is extensive, of high quality, and encompasses nearly the entire distribution range of the beech, I find the analysis performed in this study is somehow very routine and the results overall lack novelty. In the following are some of my suggestions that the authors could consider to improve the ms. The biggest issue I am afraid is the lack of deep- and novel- analyses and also lack of validation of the results. While this study explores the environmental adaptation mechanisms of European beech, it is somehow too superficial and should consider to include more in-depth investigations into functional mechanisms. On the one hand, the authors needs to explore the environmental-associated genes and variants and some molecular experimental validation is highly needed. On the other hand, as a large collection of European beech materials sourced from various locations are cultivated in a homogeneous garden and growth traits are available from previous studies, another part that could be done more is to incorporate the common garden trait results with the genomic prediction results to compare their consistency and differences, and to check the reason and mechanism behind it. Overall, I think the authors should deeply explore the dataset and do much more than the routine analyses performed in most studies, for example, the effects of gene flow, the deleterious mutation load, migration, plasticity on the genomic prediction to future climate change.

In addition to the above main point, some other comments are below:

(1) For the population structure analysis, the authors relied solely on the results of PCA to evaluate the population structure. The analysis like ADMIXTURE is also needed to examine per-individual ancestor composition. In addition, the relative role of geography, environmental and their interaction is needed to examine their contribution to neutral and adaptive genomic variation in European beech. Based on the results of PCA (Fig 1A), it seems there is extensive gene flow and hybrid clines among different beech populations. I think that incorporating considerations of gene flow in the study of local adaptation and future predictions would greatly benefit the article.

(2) It solely utilized LD-pruned variants to conduct the environmental association analysis. I am worried that this approach may diminish the comprehensiveness and power of LFMM in detecting candidate variants. Additionally, I believe that utilizing only a small subset of the dataset does not fully exploit the advantages of whole-genome data. In addition, researchers were able to identify as many as 20k adaptive variants from a pool of only 540k variants through the B-test, and the Manhattan plot reveals a substantial number of highly significant adaptive variants ($-\log P > 30$). I am particularly concerned about potential false positives and would recommend that the researchers perform accuracy validation to address this issue (i.e. IBE and so on). The detection of environmentally associated variants is crucial for RONA predictions, and therefore, some alternative methods (i.e. RDA) for environmental association detection may need to validate the results of LFMM. And the authors might need to use more climate change

scenarios and models to robust the results of genetic gap.

(3) The analytical methods employed in this study lack innovation, and the visual presentation of the figures is not aesthetically pleasing. I suggest integrating the maps (Fig2b and Fig3) with environmental variables to display in the manuscript. There is room for improvement in the writing of this manuscript as well. In addition, L77, L79, L97, L208—The ‘p’ and ‘r’ need to be italic.

REVIEWER COMMENTS

Reviewer #1 (Remarks to the Author):

Dear editor, dear authors,

In this manuscript, the authors explore the genomic bases of local adaptation to climate in a European tree species – *Fagus sylvatica* – whole genome resequencing data for 865 individuals. After identifying a set of putative adaptive variants, they estimate a risk of maladaptation to climate change across the sampled populations. I believe the study to have great potential, but all the work is based on the assumed presence of local adaptation to climate in the studied populations of *Fagus sylvatica*, when the results of a large common garden and the present genomic results could suggest the opposite. If the authors do not provide evidence of local adaptation to climate in their sampling first, the study focus on genomic offset could become questionable.

Thank you very much for raising this important point. In fact, we simply assumed that local populations will be locally adapted, as is done in many papers on many different plant and animal species. However, since we actually have the possibility to experimentally test for signals of local adaptation, which is not possible for many species, we completely agree that we need to take advantage of this opportunity. We have therefore added data from a second common garden allowing us to analyze a reciprocal transplant experiment. As detailed below, these data suggest extensive phenotypic plasticity overlaying possible adaptive differentiation. Nevertheless, we also highlight a genomic signal of polygenic adaptation that might be important under more natural conditions. In any case, we have completely changed the focus of the manuscript according to this and other additional analyses. The title of our paper now highlights “phenotypic plasticity” instead of “future maladaptation”.

My comments are listed below.

Global comments:

*1) Genomic offsets are pertinent only if the species exhibit local adaptation to climate. The manuscript should convince the reader that it is the case for *F. sylvatica*. For example, what do the common garden results tell us? Has it been analyzed already in another study that the authors could refer the reader to? The last paragraph of the discussion seems to suggest that there are only very weak signs of local adaptation among the studied populations.*

There are several studies claiming local adaptation in beech, which we have now added in the introduction (L66-67: “Accordingly, evidence of local adaptation has been reported at different geographical scales^{25,26}, although phenotypic plasticity appears pervasive^{27,28}.”) At the same time, previous analyses already reported the importance of phenotypic plasticity.

Our new analyses indicate that there may be local adaptation to climate. However, the extent appears weak to non-existent under our common garden settings. More importantly, we now provide a much more comprehensive analysis of genotype-environment-associations. While we lose almost all previously reported significant associations, we still identify evidence of polygenic adaptation for some environmental variables. These may or may not translate into performance and fitness differences in natural ecosystems (with competition by other species and rare weather extremes). In any case, adaptive differentiation is absolutely minimal in an artificial common garden setting. For this and other reasons, genomic offset measures can only be interpreted with extreme caution and should not be used as estimates for future climate vulnerability. Nevertheless, the

distribution of potentially adaptive variation in relation to climate across the landscape provides interesting insights.

*2) Assuming that the first point is resolved, the most interesting part of the study for me is the part where the authors explored the genomic bases of local adaptation to climate, including the polygenic nature of it, the potential link with negative selection, the key environmental drivers of adaptation, the genes identified, the consistency across different geographic scales... I would suggest shifting the focus of the manuscript towards these particular questions, that are still unanswered for *F. sylvatica* and even for a lot of tree species. That would mean conducting supplementary analyses (genetic variance partitioning in a multivariate way, GO terms enrichment, comparing results based on the geographic scale used...) but would make the study very interesting.*

Thank you for these useful suggestions. Indeed, we have shifted the focus of our manuscript towards a better understanding of the genetic basis of local adaptation in beech. We have extended the general population genomics analyses, looking in more detail into the three ancestry groups identified in our data. We used this information to define local metapopulations, to analyze GEAs across different geographic scales, as suggested and as previously shown to be potentially useful for genome-wide association analyses (Lopez-Arboleda et al. 2021, DOI: 10.1093/molbev/msab208). We now specifically analyze the drivers of potential local adaptation using lfm analyses on randomized data and the gradient forests method (Supplementary Figs. 14 and 15). Most importantly, by randomizing geographic coordinates of our populations and re-running the GEA analyses on these, we identify an overwhelming signal of non-causal associations. We use this to define a false-discovery rate (FDR) threshold which highlights one single high-confidence associated region, which we characterize in detail, potentially important for winter temperature adaptation acting by modulating spring phenology.

Comparison of real vs. random data for the individual environmental variables indicates a polygenic signal for some of them (especially isothermality, bio3). Considering our small sample size (compared to well-powered GWAS studies) together with the expected high polygenicity of adaptive traits, it may not be surprising that most of the genomic signal of local adaptation remains unresolved. This missing heritability, however, leads to relatively few associations which precludes GO terms enrichment analysis. Also, when comparing real to random associations, variant annotation does not indicate negative selection anymore.

In conclusion, our study now highlights the complexity of identifying the genomic basis of local adaptation and the need to correct for non-causal signal, which we believe to be an important finding for the community.

*3) Showing the attempt of validating genomic offset predictions is necessary, even if it failed. However, to me, it could suggest here again that local adaptation to climate is too weak the studied populations of *F. sylvatica* to conduct this type of analyses.*

We now show the validation analysis (in Fig. 5f). However, adaptive differentiation in the common gardens is in fact too weak to draw any firm conclusions. At the same time, the lack of differentiation under artificial settings does not exclude a role of adaptive variants in natural ecosystems.

4) Wording around the genomic offset metric in the introduction can sometimes be misleading: (Line 26) "estimates of species' responses to climate change"; (Line 28) "account for dynamic population genetic processes". Genomic offset does not predict a future intrinsic change in the population but only the amount of climate shift that will affect the current gene ~ climate relationship. Species or populations' responses are not that easily predictable. This response, plastic change? decline?

adaptation? extinction? migration? will depend on a lot of other factors that are not considered here. I believe it is critical information that must be carefully stated, especially early on in the introduction.

Thank you for pointing this out. We tried to improve our wording. In general, we moved the focus away from genomic offset. We still write that “there is an urgent need for reliable estimates of species’ responses to climate change” (now L35-36). However, we tried to more carefully state the possibilities to provide such predictions. We have deleted the “dynamic population genetic processes” sentence, as it was in fact misleading. Instead we now focus on the relevance of “adaptive differentiation and adaptive capacity of local populations” and cite the paper by Aguirre-Liguori et al. 2021, which mentions the additional evolutionary processes that might be considered (such as gene flow, migration and genetic load), which we pick up again later in the manuscript.

Specific comments:

Line 26: Neither genomic offset nor SDM can actually “estimate species’ responses to climate change”. These metrics/models just predict where the genes ~ climate (genomic offset) or the presence ~ climate (sdm) relationship will be the most affected by a predicted change in climate. In other words, it estimates where species would have to respond the most to remain present or locally adapted, but never the response itself.

This is a valid point. Of course, this is a simplification, but we feel that we don’t have the space here for a more comprehensive discussion. We believe that “there is an urgent need for reliable estimates of species’ responses to climate change.” Often SDMs or genomic offset measures are used to infer possible species’ response to climate change, even if those measures actually estimate where species would have to respond the most to remain present or locally adapted. We have changed the text slightly to now read that “Such estimates can be derived from...” We hope that the “can be derived” makes clear that this is not actually a direct measure. Additionally, we have added the citation Aguirre-Liguori et al. 2021 (The evolutionary genomics of species’ responses to climate change), which discusses the topic in much more detail.

Lines 29-31: Integration of data or integration of concepts? These two last sentences of the paragraph are not very clear, consider revision.

We revised the sentence, which now reads: “The increasing feasibility of combining reciprocal transplant and common gardens experiments with population genomic data to elucidate the genetic basis of adaptive differentiation calls for an integration of intraspecific genetic variation into conservation biology frameworks¹¹⁻¹³.”

Line 45: Consider splitting that sentence.

Split as suggested.

Line 53: Not sure what the authors mean when they say: “we set out to characterize...”. Maybe “we aimed at characterizing...” or “we sought to characterize...”?

Changed as suggested (“we aimed to characterize...”).

Introduction: The Introduction mentions only very briefly some important concepts and ideas such as “maladaptation”; “genomic vulnerability”; “potential natural vegetation (PNG)”. I think it would be important to contextualize those a bit more.

Thank you for this valid point. We have markedly rearranged the introduction on “maladaptation” and “genomic vulnerability” to present a more balanced view. At the same time, we feel that for a

more comprehensive contextualization a dedicated review paper would be needed. Importantly, the new focus of our manuscript is not on genomic offset anymore. We therefore hope that the revised introduction provides sufficient contextualization.

Line 76: The authors state “remarkable correlation between the first two principal components (PCs) and geography” but only discuss PC1 in the following section, is PC2 correlated with latitude?

Thanks for pointing this out. We added a sentence and figure parts (Fig. 1c and e) showing the correlation of PC2 with latitude.

Line 78: What are the levels of genetic differentiation among populations? It should be mentioned here.

We added an entire paragraph and figure (Fig. 2) showing different population genetic analyses, including genetic differentiation.

Line 80: I am not sure to understand how “these results demonstrate a high level of accuracy of the common garden”?

Thank you for pointing this out. We extended the sentence to better explain what we meant here: “Together, these results demonstrate a high level of accuracy of the common garden used for our analyses, as any mistakes in the organization or planting of the seeds and seedlings would have resulted in outliers or an overall erosion of the geographic signal in the genomic data.”

Figure 1: Panel b) and c) are redundant with panel a), and even potentially misleading because longitude is only relevant to PC1 (probably latitude to PC2?). The real underlying factor seems to be distance, as shown by supplementary figure 5, which would be much more appropriate in the main text, maybe as a panel of Figure 1. It is super interesting results.

Thanks for this suggestion. We added the isolation-by-distance result into the main text (Fig. 2d). Additionally, in Fig. 1 we now show detailed analyses for PC1 and PC2 (see above).

Lines 84-85: What are the results sustaining the statement “Considering the large amplitude of environmental differences, relatively strong selection for adaptive differentiation can be expected”?

This statement is not based on empirical results. We have therefore toned it down to “may be expected”.

Lines 86-87: The link between IBD (the results of the paragraph) and local adaptation is not clear to me here. Do the authors mean that populations are exchanging genetic material and must then have reached their adaptive equilibrium? That adaptive allele should exhibit frequency clines?

The link was meant between autochthony (which is related to IBD) and local adaptation. If genetic material was mixed by human activity, genotype-environment associations cannot be analyzed. To make this clearer we now write: “The autochthony of the populations together with the broad environmental range open the exciting possibility of identifying genotype-environment associations (GEAs) that can provide insights on the genetic basis of local adaptation.”

Line 110-114: The link with negative selection is quite speculative but also potentially one of the most interesting findings of that study. I would recommend adding this result to the main text and exploring further this issue, for example backing it up with a real explanation of how purifying/negative selection could affect polygenicity and adaptive allele effect sizes in a context of local adaptation.

Unfortunately, with the much more stringent definition of genotype-environment associations, the link with negative selection disappeared. We therefore had to delete this part. Nevertheless, we believe that with the new focus and the detailed analyses of GEAs and common garden data this point will not be missed as the paper should now be much more interesting in general.

Line 186-188: Because the results are only statistical predictions with no validation, I don't think the authors can say that their results "demonstrate the potential for employing putatively adaptive variation to predict general adaptive capacity to future climate conditions..."

We agree and deleted this part.

Line 189-191: I don't see what in the results actually suggest that each forest stand should be assessed individually. How realistic would be to conduct a genomic study like this one on all important forest stands?

We changed the sentence and added a citation: "Finally, for a reliable view of the adaptive genomic variation, spatial resolution may be critical as a continuous distribution of adaptive alleles and interpolation may not be warranted²⁰." The lack of correct interpolation was shown by cross-validation experiments by Lind et al. 2023 and is suggested by the fine-scale patterns of potentially adaptive variation shown by our data which is now better highlighted ("Most importantly, our analyses highlight geographically fine-scale variation..."). We also added a sentence regarding the feasibility: "Given that the selection of loci analyzed may not actually play much of a role¹⁵, this should become increasingly feasible. Large genetic biodiversity monitoring projects are already performing high-resolution sampling of different plant and animal species^{48,49}."

Line 193-195: How can the authors predict anything in their study if the link between genetic and environmental variation is too weak?

We deleted this sentence. Nevertheless, a strong genomic signal of environmental association (when corrected for confounding variables) may indicate a role for adaptation even if phenotypic differentiation cannot be detected in experimental plots.

Lines 203-204: Genomic offset validation usually works fine even if the common garden is within the current climatic range of the species, see (Fitzpatrick et al., 2021; Capblancq & Forester, 2021; Lachmuth et al., 2023). Genomic offset is in fact very similar to the climate transfer distance approach, which has worked quite well for seed sourcing for a long time and for a lot of different tree species.

Thank you for this interesting input. Interestingly, climate transfer distance does not appear to play virtually any role for beech growth under common garden conditions. In any case, we now discuss this more thoroughly and also point out the possible importance of differentiating between artificial and natural systems and thus between a silvicultural and ecological focus. Our reciprocal transplant experiment that we now present in the new Fig. 4 may be especially interesting in this context.

Line 186-213: The information given in the last paragraphs of the discussion are troubling, to me, they suggest that local adaptation is too weak in the studied populations to conduct such a study.

We are really thankful that this point was highlighted. We are just so used to assume local adaptation that we forgot to properly analyze it empirically. As discussed above, we add several new analyses addressing this point and added relevant discussion. It will be exciting to see future studies assessing the amount of adaptive differentiation for different species under different climates and link those with genome-wide analyses.

Lines 307-308: What threshold did the authors use to consider a locus as an outlier? Many “outliers” seem to still be in the overall genomic envelope in the Manhattan plots, making me wonder if the threshold value is appropriate. I would suggest comparing a False Discovery Rate approach with q-values to the Bonferroni correction. Just to make sure that many False Positives are not influencing the signal here. What about Q-Qplots and histograms of p-values provided by LFMM? These are usually very informative.

Thank you for this important point. The p-value distributions and Q-Q plots looked just right. Nevertheless, randomization of geographic coordinates revealed an overwhelming statistically significant but non-causal signal. With this result, it is quite troubling to see so many studies that do not perform any randomization. We also find the same amount of random signal with other methods, which we have added in our revision, such as BayPass or WZA. Also, IBE is significant for associated variants in a partial Mantel test with real and random data. Considering that genotype-environment association analyses are conceptually similar to genome-wide association studies, only with unmeasured fitness phenotypes approximated by the environment instead of direct phenotypic measurements, and given the expected polygenicity of adaptive traits, most GEA studies are probably severely underpowered. We believe that some kind of permutation test is essential for any GEA analysis.

Thanks for the opportunity to review this manuscript. I am sure that the study could make an important contribution to the field once its foundations are strengthened.

Thibaut Capblancq

We genuinely thank you for your time to review and comment on our manuscript. Your input helped us a lot to improve our study. Instead of a standard landscape genomics paper, we now present a much more complete and more interesting story that we hope to be important for a broader community.

Literature cited:

10. Aguirre-Liguori, J.A., Ramírez-Barahona, S. & Gaut, B.S. The evolutionary genomics of species' responses to climate change. *Nature Ecology & Evolution* **5**, 1350-1360 (2021).
11. Sork, V.L. et al. Putting the landscape into the genomics of trees: approaches for understanding local adaptation and population responses to changing climate. *Tree Genetics & Genomes* **9**, 901-911 (2013).
12. Flanagan, S.P., Forester, B.R., Latch, E.K., Aitken, S.N. & Hoban, S. Guidelines for planning genomic assessment and monitoring of locally adaptive variation to inform species conservation. *Evolutionary Applications* **11**, 1035-1052 (2018).
13. Waldvogel, A.-M. et al. Evolutionary genomics can improve prediction of species' responses to climate change. *Evolution Letters* **4**, 4-18 (2020).
15. Fitzpatrick, M.C., Chhatre, V.E., Soolanayakanahally, R.Y. & Keller, S.R. Experimental support for genomic prediction of climate maladaptation using the machine learning approach Gradient Forests. *Mol Ecol Resour* **21**, 2749-2765 (2021).
20. Lind, B.M. et al. How useful is genomic data for predicting maladaptation to future climate? *bioRxiv*, 2023.02.10.528022 (2023).
25. Kreyling, J. et al. Local adaptations to frost in marginal and central populations of the dominant forest tree *Fagus sylvatica* L. as affected by temperature and extreme drought in common garden experiments. *Ecology and Evolution* **4**, 594-605 (2014).
26. Gauzere, J., Klein, E.K., Brendel, O., Davi, H. & Oddou-Muratorio, S. Microgeographic adaptation and the effect of pollen flow on the adaptive potential of a temperate tree species. *New Phytologist* **227**, 641-653 (2020).

27. Gárate-Escamilla, H., Hampe, A., Vizcaíno-Palomar, N., Robson, T.M. & Benito Garzón, M. Range-wide variation in local adaptation and phenotypic plasticity of fitness-related traits in *Fagus sylvatica* and their implications under climate change. *Global Ecology and Biogeography* **28**, 1336-1350 (2019).
28. Kurjak, D. *et al.* Inter-provenance variability and phenotypic plasticity of wood and leaf traits related to hydraulic safety and efficiency in seven European beech (*Fagus sylvatica* L.) provenances differing in yield. *Annals of Forest Science* **81**, 11 (2024).
42. Lopez-Arboleda, W.A., Reinert, S., Nordborg, M. & Korte, A. Global Genetic Heterogeneity in Adaptive Traits. *Molecular Biology and Evolution* **38**, 4822-4831 (2021).
48. Pearman, P.B. *et al.* Monitoring of species' genetic diversity in Europe varies greatly and overlooks potential climate change impacts. *Nature Ecology & Evolution* **8**, 267-281 (2024).
49. Hoban, S. *et al.* Global genetic diversity status and trends: towards a suite of Essential Biodiversity Variables (EBVs) for genetic composition. *Biological Reviews* **97**, 1511-1538 (2022).

Reviewer #2 (Remarks to the Author):

*This study focused on the European key forest tree species, European beech (*Fagus sylvatica*), and reported a total of 653 whole-genome re-sequencing novel data, collected from 98 populations covering its entire distribution range. By employing population genetics analysis, environmental association analysis, and genetic offset prediction, the study assessed beech's landscape of future climate maladaptation risks and proposes recommendations for conservation strategies. Although the re-sequencing data used in this study is extensive, of high quality, and encompasses nearly the entire distribution range of the beech, I find the analysis performed in this study is somehow very routine and the results overall lack novelty. In the following are some of my suggestions that the authors could consider to improve the ms.*

Thank you for your evaluation of our manuscript and for highlighting the quality of our data. We agree that the initial version of our paper lacked conceptual novelty. We have performed extensive additional analyses and believe that the revision now clearly goes beyond the routinely performed landscape genomics analyses and will be more interesting to a broader audience.

The biggest issue I am afraid is the lack of deep- and novel- analyses and also lack of validation of the results. While this study explores the environmental adaptation mechanisms of European beech, it is somehow too superficial and should consider to include more in-depth investigations into functional mechanisms.

After continued work on our data, we agree that the initial analyses lacked novelty. We have thus strongly extended the exploration of the environmental adaptation in beech by many additional analyses of the genomic data (detailed below) but also by adding data from a second common garden for analyzing a reciprocal transplant experiment.

On the one hand, the authors needs to explore the environmental-associated genes and variants and some molecular experimental validation is highly needed.

Thank you for this comment. After correcting for non-causal associations by using the results of randomized data, we identify a single high-confidence genomic region (new Fig. 3). We now go much more into detail regarding the characterization of this locus. We analyze LD-patterns, compare the two haplotypes by *de novo* genome assemblies, show an effect on spring phenology and analyze expression of a candidate gene by qRT-PCR of winter buds sampled in December. We believe that these additional molecular experiments provide interesting biological insights.

On the other hand, as a large collection of European beech materials sourced from various locations are cultivated in a homogeneous garden and growth traits are available from previous studies, another part that could be done more is to incorporate the common garden trait results with the genomic prediction results to compare their consistency and differences, and to check the reason and mechanism behind it.

This is a very useful suggestion. As mentioned above, we did incorporate trait data (spring bud burst and growth) from the common garden to understand the potential mechanism underlying the most significantly associated genomic region possibly important for adaptation to different winter temperatures. Additionally, we have added growth and mortality data from a second common garden representing a different climate, again to try to better understand the biology behind the GEA results.

Overall, I think the authors should deeply explore the dataset and do much more than the routine analyses performed in most studies, for example, the effects of gene flow, the deleterious mutation load, migration, plasticity on the genomic prediction to future climate change.

Thank you for this suggestion. We agree that the analysis of gene flow may give interesting new insights. Still, we have decided to leave that for an independent future study. Nevertheless, we did calculate additive and recessive genetic load and compared it to genomic offset estimates (Fig. 5e).

In addition to the above main point, some other comments are below:

(1) For the population structure analysis, the authors relied solely on the results of PCA to evaluate the population structure. The analysis like ADMIXTURE is also needed to examine per-individual ancestor composition. In addition, the relative role of geography, environmental and their interaction is needed to examine their contribution to neutral and adaptive genomic variation in European beech. Based on the results of PCA (Fig 1A), it seems there is extensive gene flow and hybrid clines among different beech populations. I think that incorporating considerations of gene flow in the study of local adaptation and future predictions would greatly benefit the article.

We agree that the analysis of ancestry is important and therefore added a new Fig. 2. We show the ancestry composition of all individuals with regard to three major ancestry clusters and show how these clusters are distributed across the range. As mentioned above, we will specifically characterize gene flow and hybridization between lineages in an independent study.

(2) It solely utilized LD-pruned variants to conduct the environmental association analysis. I am worried that this approach may diminish the comprehensiveness and power of LFMM in detecting candidate variants. Additionally, I believe that utilizing only a small subset of the dataset does not fully exploit the advantages of whole-genome data. In addition, researchers were able to identify as many as 20k adaptive variants from a pool of only 540k variants through the B-test, and the Manhattan plot reveals a substantial number of highly significant adaptive variants ($-\log P > 30$). I am particularly concerned about potential false positives and would recommend that the researchers perform accuracy validation to address this issue (i.e. IBE and so on). The detection of environmentally associated variants is crucial for RONA predictions, and therefore, some alternative methods (i.e. RDA) for environmental association detection may need to validate the results of LFMM. And the authors might need to use more climate change scenarios and models to robust the results of genetic gap.

We did perform the LFMM analysis with the non-LD-pruned dataset with similar results. Since in the end we identify a single highly associated region representing an extended region of suppressed

recombination, for whose analysis we use the non-LD-pruned data, the 540k variants give us enough resolution to delineate the region.

We comprehensively addressed the point of potential false positives also mentioned by Reviewer #1 (see above). Thank you for pointing this out as well. The amount of significant but non-causal signal from randomized data was completely surprising to us, considering that several recent papers simply report GEA results without any threshold correction by randomization. This random signal also occurs using other methods, such as BayPass or WZA (Supplementary Fig. 9). Notably, these alternative methods support the high-confidence region identified by our initial LFMM analysis (Fig. 3b-d).

Since we moved the focus away from genomic offset and mainly use that to highlight the spatial resolution of potentially adaptive variation, we did not include additional climate change scenarios or models. Nevertheless, we did include a gradient forests analysis.

(3) The analytical methods employed in this study lack innovation, and the visual presentation of the figures is not aesthetically pleasing. I suggest integrating the maps (Fig2b and Fig3) with environmental variables to display in the manuscript. There is room for improvement in the writing of this manuscript as well. In addition, L77, L79, L97, L208—The ‘p’ and ‘r’ need to be italic.

Thank you for this critical assessment. We changed p and r to italics. Also, we added several new data analyses and largely rewrote the article. We believe that the manuscript now is much more interesting and relevant. Also, based on your feedback, we revised the figures paying special attention to a more aesthetic visual representation.

We would also like to thank you for your time to review our manuscript. We found the comments really helpful to improve the quality of our paper.

REVIEWER COMMENTS

Reviewer #1 (Remarks to the Author):

Dear editor, dear authors,

This is a manuscript I reviewed before. In this study, the authors explore the genomics of *Fagus sylvatica* across the European climatic landscape and look for the genomic basis of local adaptation in this forest tree species. While the authors partially refocused the study according to my and the two other reviewers' comments, this new version of the manuscript seems unachieved. I listed below my concerns, questions and suggestions.

Global comments:

1) The introduction remains entirely focused on predicting the impact of climate change. I would suggest working on refocusing at least one paragraph towards the interest of better understanding the genomic basis of local adaptation to climate, which is a requisite before integrating genomic data into predictive models.

2) I wonder if, as proposed by the authors, local adaptation to climate is actually polygenic in *Fagus sylvatica*? My read on the authors' results is more that there is one genomic region that shows signals of selection for climate factors and that this region could host only a few genes linked to one unique trait: bud break phenology. Or maybe the threshold to identify genomic regions linked to selection is not appropriate (see below)?

3) The authors have the data, so why not doing a genotype~phenotype analysis for bud flush in the common garden? This could nicely back up what was found by the GEA and the link with bud flush phenology.

4) The randomization analysis conducted by the authors to lower the number of false positives is potentially interesting, but I have a few concerns and questions:

a. First, the way it is presented can be misleading, I initially believed that the authors found a way to correct their GEA procedures to identify causal variants (see abstract). But, if I'm correct, they just used a more complicated way to find the best threshold value for considering that a GEA is significant or not.

b. Their procedure is almost not explained in the methods, keeping the reader from understanding what was done.

c. I am not sure we can call this procedure a false discovery rate (FDR) analysis. The authors produced what I would call null distributions of test statistics/p-values and compare them to their observed distribution. By randomizing the sites and genotypes they produce distributions that should be expected if there was no GEA signal. I was expecting them to then project their observed distribution on these null distributions to estimate p-values (as z-scores) in a more accurate way. If I'm correct, that is not what they did but that's where the Methods are not sufficient to be sure. It seems that they assume that the 5th and 20th percentiles of these null distributions to inform on p-value thresholds below which we can expect no more than 5 or 20 percent of false positives. If that's the case, I doubt it is the most appropriate procedure. I therefore wonder why the authors did not use a classical FDR approach such as the one

implemented in the q-value R-package, which is commonly used in GEA studies.

Specific comments:

Abstract: I don't think the two citations of the abstract are necessary.

Line 15: I would not use "non-causal" here or throughout the manuscript to characterize putative false positives. These loci are just not significant according to the authors decision threshold.

Lines 18-19: Consider replacing "the highest confidence genomic region" by "the one genomic region that constantly shows strong signal of selection".

Lines 20-21: This part of the abstract sounds like phenology and growth should be correlated, but is that always an expectation?

Lines 156-158: The IBE signal could be due to very high correlation between geographic and environmental variation. Was this possibility considered? It is not totally clear from the Methods what was done here. As already suggested in my previous comments, variance partitioning could give us the proportion of genetic variance really explained by climate factors only, which is probably very low.

Line 193: "which is again in line with the polygenic adaptation theory". Not sure to follow the authors' logic here.

Lines 194-196: Based on what I have understood of conifers' phenological adaptations when I was working on spruce, populations in cooler environments need to accumulate more degree days above zero before flushing than the ones in colder environments (high latitudes and/or elevation especially). Counter intuitively, that makes cold-adapted genotypes flushing earlier in common gardens with mild climates than other genotypes. It is indeed linked to frost avoidance in milder environments where trees can accumulate a lot of degree days very early in the season. But it is not linked to the actual timing of the frost events during the year. Late frosts are probably happening much earlier in the year near the Atlantic than in eastern or northern Europe.

Line 239: Difficult to fully understand what was done in Supplementary figure 14 but we do not know if these associated loci are scattered on the genome or packed on chromosome 2. I would expect that a large part of chromosome 2 is affected by the action of divergent selection, driving many loci to show association with climate, even if they are not themselves associated with the environment.

Line 261-262: It is not clear to me why the authors expected a correlation between genetic load and genomic offset.

Line 289: A R^2 of 0.9% does not corroborate anything unfortunately, but even if it was higher I don't understand how it would corroborate a "polygenic" adaptation.

Figure 5: Change population size to sample size.

I really believe that this study has potential to become a strong reference for future genomic analyses in beech and more largely in tree species and I'm therefore looking forward to seeing how the authors will improve their manuscript.

Thibaut Capblancq

Reviewer #2 (Remarks to the Author):

In the revised manuscript, I acknowledge the authors' efforts to enhance it by incorporating comments and suggestions from the two reviewers, particularly by including the results of reciprocal transplant experiments, which bolster the novelty of the study. However, I have identified several serious issues with the logical coherence of the narrative and the alignment of results in this study. Some sections appear disjointed, and certain results seem to contradict each other. Additionally, there are concerns regarding the inference and conclusion parts that require further refinement. Below are the main issues that need attention:

1. I recommend separating the results and discussion sections, with distinct headings for different subsections within the results to emphasize the key findings of each section.

2. The primary issue lies in the logical coherence of the study's narrative. The overarching aim is to investigate the genetic basis of local adaptation in European beech populations across their range and to predict their genetic offset and maladaptation for forest conservation amidst climate change. However, apart from the population structure analysis, the results only yield one locus associated with a single environmental variable post-randomization correction, indicating a highly polygenic nature of local adaptation. While randomization correction reduces false-positive signals, it may increase false negatives, particularly when there's a high alignment between population genetic structure and local environmental adaptation. Given the polygenic architecture, the authors employ three approaches (LFMM, BayPass, and WZA) that mainly assume oligogenic and clinal adaptation patterns. I suggest considering an approach more suitable for polygenic adaptation, such as RDA, which could also aid in predicting genetic offsets.

3. The presence of common garden and reciprocal transplant experiments in this system is advantageous. However, it is surprising to observe extensive phenotypic plasticity in the two selected traits (stem diameter and survival), which somewhat contradicts the main aim of the study to detect the genetic basis of local adaptation. Additionally, there seems to be a logical gap between the results presented in Figure 4 and those before (Figure 3) and after (Figure 5). Since the authors have trait data from two common gardens, it would be beneficial to perform GWAS analysis for the two target traits and identify potential loci associated with them.

Furthermore, the trait values used here are observed rather than genetic, suggesting a need to distinguish between genetic, environmental, and their interaction effects on the target traits, especially considering the availability of reciprocal transplant experiment data. With this, the authors can compare the BLUP genetic breeding values of the target traits and assess their association with any of the 19 environmental variables across all individuals. Moreover, for Figure 5, it is recommended that the authors compare the association between BLUP values and genetic offset (RONA, genetic offset derived from gradient forest or RDA approach).

4. Apart from stem diameter and survival traits, considering phenology-related traits like bud

flush and bud set, which are often associated with local adaptation in trees, could be valuable.

5. There appears to be an issue regarding the estimation of genetic load, as it should be relative to synonymous variants to adjust for heterozygosity differences across individuals. Additionally, to evaluate genetic load across individuals, all polymorphic sites rather than only LD-pruned unlinked sites should be considered.
6. Given the polygenic nature of local adaptation, it would be intriguing to compare the RDA-predicted trait values (as proposed by Lotterhos, PNAS, 2023) with the traits measured in common gardens, as well as the genetic offset to future climate change.
7. The study currently utilizes the MPI-ESM1-2-HR model exclusively, along with different shared socioeconomic pathways. However, I recommend that the authors incorporate additional models from Worldclim and demonstrate the variation in estimated offsets and RONA values

Additionally, there are several other issues as follows:

1. Line 186-187: The statement "a natural gene variant" lacks supporting evidence.
2. Line 210-212: The assertion that "allelic variation of the genomic region did not affect stem circumference" lacks appropriate justification. The same applies to lines 212-213.
3. Line 220-222: The statement "Given the substantial genetic differentiation" is problematic considering the low genetic differentiation observed from F_{st} estimation. This explanation requires clarification.
4. Line 243-245: The inference that "the many associated...climate change in principle" is not appropriate and requires revision.
5. Line 268-270: The assertion "The patterns do not appear to be caused by the distribution of predicted climate change" contradicts the observation from supplementary Figure 17, where the distribution of predicted climate change is highly correlated with genetic offset estimation. This contradiction needs to be addressed.
6. Filtering linked loci for GEA analysis does not guarantee that identified outliers are major effect loci. Therefore, it is still recommended to use all polymorphic variants (maybe only filter minor allele frequency).
7. Regarding Figure 3h, it is recommended to validate structural variation by incorporating resequencing data. Selecting individuals from different taxonomic groups and using tools like *Samplot* can confirm whether structural variation is fixed in different ecotypes.
8. The RONA analysis in the manuscript only highlights results for bio3, which may be unreasonable. It is recommended to include RONA analysis for rainfall variables as well, given the significance of rainfall for plant growth. Additionally, since the GF analysis employs 19 environmental variables, the RONA analysis should also consider multiple variables instead of solely focusing on temperature variables. A comprehensive assessment of a species' future maladaptation risks is essential for developing targeted conservation policies.

REVIEWER COMMENTS

Reviewer #1 (Remarks to the Author):

Dear editor, dear authors,

*This is a manuscript I reviewed before. In this study, the authors explore the genomics of *Fagus sylvatica* across the European climatic landscape and look for the genomic basis of local adaptation in this forest tree species. While the authors partially refocused the study according to my and the two other reviewers' comments, this new version of the manuscript seems unachieved. I listed below my concerns, questions and suggestions.*

Thank you very much for reviewing the revised version of our manuscript and again providing helpful comments and suggestions. We hope that this further revision solved the remaining issues.

Global comments:

1) The introduction remains entirely focused on predicting the impact of climate change. I would suggest working on refocusing at least one paragraph towards the interest of better understanding the genomic basis of local adaptation to climate, which is a requisite before integrating genomic data into predictive models.

We agree that the introduction was still focused too much on predicting the impact of climate change which does not properly reflect the focus of our study. As suggested, we have added a paragraph on the elucidation of the genetic basis of local adaptation (lines 43-51), which is an important part of our manuscript, even more so in this new revision.

*2) I wonder if, as proposed by the authors, local adaptation to climate is actually polygenic in *Fagus sylvatica*? My read on the authors' results is more that there is one genomic region that shows signals of selection for climate factors and that this region could host only a few genes linked to one unique trait: bud break phenology. Or maybe the threshold to identify genomic regions linked to selection is not appropriate (see below)?*

Thank you for highlighting the insufficient clarity of our presentation regarding the genomic architecture of local adaptation in beech. We have rewritten several parts of the GEA analyses to better explain the polygenic nature of local adaptation in any species and specifically in beech. Rather than resulting from a threshold issue, we believe (and provide new supporting data) that we are dealing with a missing heritability issue due to the highly polygenic nature of complex adaptive traits. The one locus is simply the only one differentiated strongly enough to be identified with confidence in our analysis, but it represents only a tiny fraction of the overall genomic basis of local adaptation in beech.

3) The authors have the data, so why not doing a genotype~phenotype analysis for bud flush in the common garden? This could nicely back up what was found by the GEA and the link with bud flush phenology.

We did not want to overload the paper, but after careful reconsideration, we found that genotype-phenotype association analyses actually represent a useful addition to our story, especially since they

more explicitly allow testing for missing heritability, which we believe is an underappreciated point in GEA studies.

We therefore now present GWAS results for bud burst and stem circumference for the same 653 beech individuals also used in our GEA analyses. We show that we can explain a substantial amount of phenotypic variation (in bud burst) with a genomic prediction model, but only very little with the significantly associated GWAS peaks (which are not consistent across years), demonstrating high levels of missing heritability and a highly polygenic architecture. Given that environmental adaptation is a composite trait and is thus even more complex than the individual fitness-associated phenotypes, such as bud burst, this result suggests even higher levels of missing heritability in GEA studies.

4) The randomization analysis conducted by the authors to lower the number of false positives is potentially interesting, but I have a few concerns and questions:

a. First, the way it is presented can be misleading, I initially believed that the authors found a way to correct their GEA procedures to identify causal variants (see abstract). But, if I'm correct, they just used a more complicated way to find the best threshold value for considering that a GEA is significant or not.

This is a valid point and we did not mean to imply that we actually found a soft filtering method truly differentiating causal from non-causal variants. Our method, as any other GEA method, still uses a hard cutoff, which cannot distinguish causal from non-causal signal. We changed several sentences to avoid any misleading statements and now more explicitly explain the methodology in the main text (lines 161-180) but also in the according part of the Supplementary Material.

b. Their procedure is almost not explained in the methods, keeping the reader from understanding what was done.

Sorry for this and thanks for pointing it out. We have extended the methods regarding our randomization and also explained our methodology and the logic behind in the main text (see answer above) and in much more detail in the according figure legends (especially Supplementary Fig. 7).

c. I am not sure we can call this procedure a false discovery rate (FDR) analysis. The authors produced what I would call null distributions of test statistics/p-values and compare them to their observed distribution. By randomizing the sites and genotypes they produce distributions that should be expected if there was no GEA signal. I was expecting them to then project their observed distribution on these null distributions to estimate p-values (as z-scores) in a more accurate way. If I'm correct, that is not what they did but that's where the Methods are not sufficient to be sure. It seems that they assume that the 5th and 20th percentiles of these null distributions to inform on p-value thresholds below which we can expect no more than 5 or 20 percent of false positives. If that's the case, I doubt it is the most appropriate procedure. I therefore wonder why the authors did not use a classical FDR approach such as the one implemented in the q-value R-package, which is commonly used in GEA studies.

We agree that it is better to not call our cutoff FDR and changed that.

We did use the classical FDR approach and the more conservative Bonferroni correction, but obtained similar results than with random data. This indicates that "running a genotype-environment association analysis and simply correcting for multiple testing may yield an exceedingly high number

of false positives, that is variants that are statistically associated but have no biological relevance.” (lines 165-167).

We did project our data onto an assumed null distribution to extract z-scores and estimate p-values, using the ‘lfmm.pvalues’ function (and similar functions when using baypass or WZA). However, this resulted in many false positives, as shown by the random data analyses.

We agree that it would be most appropriate to use the random p -value distributions and project the p -values from the real data onto this distribution. Conceptionally (but not formally), our determination of a randomization-corrected significance threshold does that. We would love to collaborate with statisticians to develop a more formal method using data randomization for significance threshold calculations. However, we lack the statistical expertise to do this ourselves and feel that this is outside the scope of the current study. Nevertheless, we now much more explicitly explain our methodology and logic behind it in the main text, the methods part and the Supplementary Material (see answers above).

Specific comments:

Abstract: I don't think the two citations of the abstract are necessary.

We removed the citations from the abstract.

Line 15: I would not use “non-causal” here or throughout the manuscript to characterize putative false positives. These loci are just not significant according to the authors decision threshold.

We completely agree that putative false-positives cannot be called non-causal and changed one misleading sentence. However, in line 18 and several other instances throughout the manuscript, we report definite false positives (as they occur in analyses of random data) and not putative false positives. We now make this explicitly clear in lines 163-164: “Since the genotypes are not associated with the random coordinates in any biologically meaningful way, this signal is non-causal by definition.”

Using common GEA methods and FDR correction, these variants are significant despite not having any biological meaning.

Lines 18-19: Consider replacing “the highest confidence genomic region” by “the one genomic region that constantly shows strong signal of selection”.

Changed as suggested.

Lines 20-21: This part of the abstract sounds like phenology and growth should be correlated, but is that always an expectation?

We think that this is not always an expectation, even though a correlation is apparent in some tree species. However, we believe that an important adaptive sequence variant, increasing fitness under specific conditions, may affect growth in beech, as growth is an important proxy for fitness. If the adaptive locus affects fitness via phenology, we would expect an effect on both traits.

To avoid any confusion, we changed the sentence in the abstract: “Surprisingly, however, allelic variation at this locus did not result in any apparent fitness differences in a common garden.”

Lines 156-158: The IBE signal could be due to very high correlation between geographic and environmental variation. Was this possibility considered? It is not totally clear from the Methods what was done here. As already suggested in my previous comments, variance partitioning could give us the proportion of genetic variance really explained by climate factors only, which is probably very low.

We did correct for geographic variation in the IBE analysis. We now mention this explicitly also in the main text: “Isolation-by-environment (IBE) of associated variants, corrected for geographic structure, ...” (lines 187-188)

Additionally, we performed a partial RDA for variance partitioning, as suggested, now presented in Supplementary Table 4. We find that, similar to other studies, about one third of the variance that can be explained by the full model (11% from 36%) can be attributed to differences in climate, that is our selected 12 bioclimatic variables.

Line 193: “which is again in line with the polygenic adaptation theory”. Not sure to follow the authors’ logic here.

We deleted this sentence.

Lines 194-196: Based on what I have understood of conifers’ phenological adaptations when I was working on spruce, populations in cooler environments need to accumulate more degree days above zero before flushing than the ones in colder environments (high latitudes and/or elevation especially). Counter intuitively, that makes cold-adapted genotypes flushing earlier in common gardens with mild climates than other genotypes. It is indeed linked to frost avoidance in milder environments where trees can accumulate a lot of degree days very early in the season. But it is not linked to the actual timing of the frost events during the year. Late frosts are probably happening much earlier in the year near the Atlantic than in eastern or northern Europe.

Thank you for pointing this out. We removed “late” and instead added “less predictable climates”. The text now reads: “Interestingly, the reference allele of the cold-associated haplotype dominantly delays spring phenology by about 2.5 days (one-way ANOVA and Tukey’s post hoc HSD test, $p < 0.001$, Fig. 3g) which could contribute to frost avoidance in the less predictable climates of the Atlantic part of the range.”

Line 239: Difficult to fully understand what was done in Supplementary figure 14 but we do not know if these associated loci are scattered on the genome or packed on chromosome 2. I would expect that a large part of chromosome 2 is affected by the action of divergent selection, driving many loci to show association with climate, even if they are not themselves associated with the environment.

This is an important point. From the Manhattan plots in Supplementary Fig. 8, we do get an overview of the distribution of these variants. For bio3, for example, there is no single dominant peak, which we now mention in the according sentence with reference to the supplementary figure. The distribution of highly associated variants indicates that they are distributed relatively evenly across the genome. More importantly, we now exactly explain what was done in the supplementary figure’s (now Supplementary Fig. 19) legend.

Line 261-262: It is not clear to me why the authors expected a correlation between genetic load and genomic offset.

We did not expect this correlation. Nevertheless, we believe that it is important to report the lack of it. We modified the sentence which now reads “Recessive genetic load was independent of genomic offset in our data highlighting another layer of complexity (Supplementary 21).”

Line 289: A R^2 of 0.9% does not corroborate anything unfortunately, but even if it was higher I don't understand how it would corroborate a “polygenic” adaptation.

We agree that the R^2 is too small to draw any robust conclusions. We therefore deleted this sentence. Also, we moved this part of the figure into the Supplementary Material to further take away focus from the genomic offset part, which we find interesting mainly because of the distribution of potentially adaptive genetic variation across the landscape.

If there was clear differentiation between the populations in our common garden and we would find a higher R^2 for growth as explained by genomic offset, to us this would indicate polygenic adaptation as we would have successfully used a random set of 10k variants to predict a fitness-related phenotype, which should only be possible if many variants across the genome contribute to this phenotype in an approximately infinitesimal way.

Figure 5: Change population size to sample size.

Done.

I really believe that this study has potential to become a strong reference for future genomic analyses in beech and more largely in tree species and I'm therefore looking forward to seeing how the authors will improve their manuscript.

Thibaut Capblancq

Thank you very much for again taking the time to help us improve our manuscript. We hope that in the newly revised version we set the focus correctly and accurately explained and synthesized our findings so that the manuscript can now serve as a strong reference for future landscape genomics studies even beyond trees.

Reviewer #2 (Remarks to the Author):

In the revised manuscript, I acknowledge the authors' efforts to enhance it by incorporating comments and suggestions from the two reviewers, particularly by including the results of reciprocal transplant experiments, which bolster the novelty of the study. However, I have identified several serious issues with the logical coherence of the narrative and the alignment of results in this study. Some sections appear disjointed, and certain results seem to contradict each other. Additionally, there are concerns regarding the inference and conclusion parts that require further refinement. Below are the main issues that need attention:

Thank you for your time to review and comment on our manuscript. Below you will find our responses to all of the comments and suggestions, which we believe contributed to further improving our manuscript.

1. I recommend separating the results and discussion sections, with distinct headings for different subsections within the results to emphasize the key findings of each section.

Thank you for this suggestion. We separated the parts and added subheadings, which we believe strongly improved the overall clarity of our manuscript.

2. The primary issue lies in the logical coherence of the study's narrative. The overarching aim is to investigate the genetic basis of local adaptation in European beech populations across their range and to predict their genetic offset and maladaptation for forest conservation amidst climate change. However, apart from the population structure analysis, the results only yield one locus associated with a single environmental variable post-randomization correction, indicating a highly polygenic nature of local adaptation. While randomization correction reduces false-positive signals, it may increase false negatives, particularly when there's a high alignment between population genetic structure and local environmental adaptation. Given the polygenic architecture, the authors employ three approaches (LFMM, BayPass, and WZA) that mainly assume oligogenic and clinal adaptation patterns. I suggest considering an approach more suitable for polygenic adaptation, such as RDA, which could also aid in predicting genetic offsets.

Thank you for this valuable comment highlighting the lack of a multivariate method especially suitable for the analysis of local adaptation in the case of highly polygenic architectures. Following this advice, we employed RDA with our data and again also with randomized data.

While we identify thousands of significant sequence variants, including our previously identified locus on chromosome 2, putatively associated with the 12 selected bioclimatic variables (which explain 11% of genetic variation in a partial RDA as mentioned above and now presented in Supplementary Table 4), the results from three RDA runs with random data show that the method may also suffer from high levels of false positives, as previously reported for simulated data by Lotterhos 2023 (10.1073/pnas.2220313120). Even though more significant GEAs are identified in the real compared to the random data, the p-value distributions are highly similar precluding the empirical determination of a significance threshold for defining high-confidence GEAs. This is shown in the new Supplementary Fig. 11.

The multivariate RDA method cannot differentiate between real and random signal either and thus cannot identify truly associated, putatively adaptive sequence variants. This notion is further supported by the lack of any significant overrepresentation of GO terms among the genes tagged by RDA-identified GEAs in our real data.

Additionally, we more specifically discuss the problem of false negatives and now present a second less stringent significance cutoff, at which additional associations are identified. Nevertheless, also taking into account simulation studies reporting high false positive rates for different GEA methods^{32,24}, we believe that the main challenge to identify the genetic basis of local adaptation is the separation of true from random signal, which may be partially possible with larger sample sizes. False negatives due to correction for population structure will, however, remain. For practical applications it may therefore be more reasonable to not attempt the genetic dissection of local

adaptation in the first place, but employ the overall genomic information instead, as can be done by using genomic prediction or genomic offset models with a representative set of random variants.

3. The presence of common garden and reciprocal transplant experiments in this system is advantageous. However, it is surprising to observe extensive phenotypic plasticity in the two selected traits (stem diameter and survival), which somewhat contradicts the main aim of the study to detect the genetic basis of local adaptation. Additionally, there seems to be a logical gap between the results presented in Figure 4 and those before (Figure 3) and after (Figure 5).

We agree with the logical gap between Figures 3, 4 and 5. In fact, we considered different orders of the results before and again now. However, we still believe that the current order allows the best coherence of the story (we tried moving the reciprocal transplant experiment forward or to the end, but both of those alternatives made the narrative worse.)

Nevertheless, the logical gap needed to be fixed. For this reason, we added transitional paragraphs, especially between the results of Fig. 4 and 5 (lines 256-266), now logically connecting the phenotypic plasticity results with the GWAS (which we newly added), which in turn connects to the genomic offset part. At the same time, we tried to reconcile the apparent contradiction between phenotypic plasticity and local adaptation, which we believe may be due to the experimental conditions.

Since the authors have trait data from two common gardens, it would be beneficial to perform GWAS analysis for the two target traits and identify potential loci associated with them.

Following this suggestion, which was also brought up by reviewer #1, we now present results from a GWAS (see answer to point #3 above for more details) presented in lines 268-283.

Furthermore, the trait values used here are observed rather than genetic, suggesting a need to distinguish between genetic, environmental, and their interaction effects on the target traits, especially considering the availability of reciprocal transplant experiment data. With this, the authors can compare the BLUP genetic breeding values of the target traits and assess their association with any of the 19 environmental variables across all individuals. Moreover, for Figure 5, it is recommended that the authors compare the association between BLUP values and genetic offset (RONA, genetic offset derived from gradient forest or RDA approach).

Thank you for this suggestion. However, since our common gardens are provenance tests, i.e. without family structure, we cannot compute BLUPs the traditional way, without genomic data. Unfortunately, we only have genomic data for one of the two common gardens (specifically the site in Northern Germany.) Therefore, we cannot use genetic breeding values for our reciprocal transplant experiment. However, we did consider (micro-)environmental effects in our linear models, by incorporating fixed effects such as block and the site, for the calculations of potential interactions between groups of provenances and sites for growth and mortality. We added previously missing information in the methods part and now report the exact model used (lines 405-409: "Analysis of variance (ANOVA) was conducted based on the following linear mixed model: $Y_{ijk} = \mu + P_i + E_j + (P \times E)_{ij} + B(E)_{jk} + e_{ijk}$, where Y_{ijk} is the phenotypic observation of a trait made for the i th provenance (P), grown at the j th environment (E), located in the k th block (B) within environment E . $P \times E$ represents the provenance by environment interaction, μ is the overall experimental mean, and e is the experimental error (residual).")

In the new GWAS part, we use BLUP genetic breeding values for genomic prediction, which works impressively well for bud burst.

Finally, we moved the part of Fig. 5 correlating phenotypes in the common garden with genomic offset estimates to the Supplementary Material to further take away the focus from the genomic offset part, which currently lacks empirical validation.

4. Apart from stem diameter and survival traits, considering phenology-related traits like bud flush and bud set, which are often associated with local adaptation in trees, could be valuable.

We agree that this would be interesting. Unfortunately, due to the work required to assess phenology (requiring either several experienced experimenters or advanced unmanned aerial vehicles on the site several times per week for about one month) we only have this data for one site at the moment. As a side note, there is no strong empirical evidence for beech so far demonstrating that the phenotypic differentiation in phenology-related traits between populations across the landscape actually translates into fitness differences. If it does, we would still expect phenology-related traits to have pleiotropic effects on other fitness-related traits, such as growth or mortality.

5. There appears to be an issue regarding the estimation of genetic load, as it should be relative to synonymous variants to adjust for heterozygosity differences across individuals. Additionally, to evaluate genetic load across individuals, all polymorphic sites rather than only LD-pruned unlinked sites should be considered.

We fully agree with the reviewer and actually the estimation of genetic load is already computed relative to synonymous variants. We have edited the text so that this is more evident to the reader (see lines 571-572 and 578). Also, as we are providing a relative estimate, we believe that it is still robust when based on only LD-pruned polymorphism.

6. Given the polygenic nature of local adaptation, it would be intriguing to compare the RDA-predicted trait values (as proposed by Lotterhos, PNAS, 2023) with the traits measured in common gardens, as well as the genetic offset to future climate change.

While we do agree that this would be an interesting analysis, we feel that this analysis is out of the scope of the current study. Nevertheless, we will consider the RDA-predicted trait values for subsequent studies also including additional analyses regarding genomic offset validation by in situ assessment of our studied populations.

Additionally, in the new part reporting the GWAS results, we at least added gBLUP-predicted bud burst phenotypes to explore the amount of phenotypic variation that we can explain using genomic prediction.

7. The study currently utilizes the MPI-ESM1-2-HR model exclusively, along with different shared socioeconomic pathways. However, I recommend that the authors incorporate additional models from Worldclim and demonstrate the variation in estimated offsets and RONA values

We focused on a single model and scenario because we did not want to give the impression that our offset values could already be used for any practical action, and if they cannot, we did not consider the variation between models to be too important. Nevertheless, it may still be interesting to assess variability caused by different climate change projections. We therefore added two additional

climate models (EC-Earth3-Veg and BCC-CSM2-MR) and one additional scenario (SSP 585 or RCP8.5). The results of all 6 projections are presented in the new Supplementary Fig. 23. Correlations of offset values using different climate models are highly significant ($p < 0.001$) and range from 0.7 to 0.8.

Additionally, there are several other issues as follows:

1. Line 186-187: The statement "a natural gene variant" lacks supporting evidence.

We meant gene in the broad sense, but changed this to "sequence variant" to not be misleading.

2. Line 210-212: The assertion that "allelic variation of the genomic region did not affect stem circumference" lacks appropriate justification. The same applies to lines 212-213.

We now show in the new Supplementary Fig. 15, that allelic variation of the most significantly linked variant, representative for the genomic region, does not affect stem circumference.

3. Line 220-222: The statement "Given the substantial genetic differentiation" is problematic considering the low genetic differentiation observed from F_{st} estimation. This explanation requires clarification.

Thank you for pointing this out. We changed "substantial" to "consistent" and added Supplementary Fig 16 to show this consistent differentiation between groups of local populations.

4. Line 243-245: The inference that "the many associated...climate change in principle" is not appropriate and requires revision.

We agree and therefore changed this sentence and moved it to the discussion: "Our GEA and GWAS analyses suggest high levels of (potentially adaptive) standing genetic variation, which might contribute to adaptability."

5. Line 268-270: The assertion "The patterns do not appear to be caused by the distribution of predicted climate change" contradicts the observation from supplementary Figure 17, where the distribution of predicted climate change is highly correlated with genetic offset estimation. This contradiction needs to be addressed.

We changed the sentence to now read "This pattern appears to be only partially explained by the distribution of predicted climate change (Supplementary Fig. 24)" and added the correlation between climate change and genomic offset into the figure, highlighting two neighboring populations from Southern Germany as an example.

6. Filtering linked loci for GEA analysis does not guarantee that identified outliers are major effect loci. Therefore, it is still recommended to use all polymorphic variants (maybe only filter minor allele frequency).

We agree that all variants, not only the LD-pruned ones, should be used for association studies to ensure that the top variant of each locus is included and to not lose any loci. However, as we show in detail by employing various methods and data randomizations, we are not at the point where we actually resolve the genetic basis of environmental adaptation. Rather, we can only identify the most significantly associated loci, in our case basically a single one on chromosome 2. On the other hand,

methods that are indifferent with respect to the specific variants used, such as genomic prediction or genomic offset, and may thus be more suitable for our data at this point should be used with LD-pruned data.

Running different GEA methods using real and random data is computationally quite demanding even with the smaller LD-pruned dataset. Nevertheless, we performed an LFMM analysis with all the 3.68 million variants before LD-pruning. The results are highly similar. Most importantly, the single high-confidence locus can again be pinpointed to the same genomic region of chromosome 2, which we now show in the new Supplementary Fig 9.

7.Regarding Figure 3h, it is recommended to validate structural variation by incorporating resequencing data. Selecting individuals from different taxonomic groups and using tools like SnpPlot can confirm whether structural variation is fixed in different ecotypes.

Thank you for this valuable suggestion. The generalization of the alignment of two individuals, one for each haplotype, was in fact an oversimplification. Unfortunately, the combination of short-read resequencing data and the collapsed consensus beech reference genome assembly (despite high levels of heterozygosity) do not allow robust inference of structural variation. We thus still believe that *de novo* assemblies employing long-read data should be used. However, this calls for a pangenome analysis which we cannot provide at this point. We therefore decided to delete this result, as it was not essential to the story.

8.The RONA analysis in the manuscript only highlights results for bio3, which may be unreasonable. It is recommended to include RONA analysis for rainfall variables as well, given the significance of rainfall for plant growth. Additionally, since the GF analysis employs 19 environmental variables, the RONA analysis should also consider multiple variables instead of solely focusing on temperature variables. A comprehensive assessment of a species' future maladaptation risks is essential for developing targeted conservation policies.

As mentioned above, we did not want to put too much emphasis on the RONA/genomic offset analyses in this manuscript. The resources described here allow for beautiful population genomics analyses and give exciting insights into the genetic basis of local adaptation using GEA and GWAS. However, with our currently analyzed common garden, which does not exhibit hardly any differentiation in growth or mortality between populations, we cannot move forward with disentangling the environmental variables most important for potential future maladaptation nor validate our offset estimates. We can only use our genomic signals to rank the importance of allelic distances. We therefore feel that a comprehensive assessment of future maladaptation of beech is not possible at this point and would thus refrain from presenting additional RONA results in the current manuscript. Nevertheless, we will certainly continue working on this and hope to have new possibilities by assessing our populations in their natural environments in the near future (as well as additional common gardens). We have added an according sentence. It will be exciting to move forward with this research.

We would like to thank reviewer #2 for the many useful comments and suggestions. We hope that the revised version of the manuscript now achieves a clear logical coherence and reconciles presumably contradicting results. We added new data, analyses and explanations to refine several inferences and conclusions. We believe that our paper provides important new insights into the genetic basis and the genomic prediction of local adaptation and hope that our results will be taken up by the landscape genomics community.

REVIEWERS' COMMENTS

Reviewer #2 (Remarks to the Author):

I appreciate the authors for addressing all my previous comments. Most of my concerns have been resolved in this version of the manuscript. Thank you.

Best wishes,
-Jing Wang